# Time-resolved single-cell sequencing identifies multiple waves of mRNA decay during the mitosis-to-G1 phase transition

Lenno Krenning[†‡], Stijn Sonneveld[†], Marvin E Tanenbaum*

Oncode Institute, Hubrecht Institute – KNAW and University Medical Center Utrecht, Utrecht, Netherlands

**Abstract** Accurate control of the cell cycle is critical for development and tissue homeostasis, and requires precisely timed expression of many genes. Cell cycle gene expression is regulated through transcriptional and translational control, as well as through regulated protein degradation. Here, we show that widespread and temporally controlled mRNA decay acts as an additional mechanism for gene expression regulation during the cell cycle in human cells. We find that two waves of mRNA decay occur sequentially during the mitosis-to-G1 phase transition, and we identify the deadenylase CNOT1 as a factor that contributes to mRNA decay during this cell cycle transition. Collectively, our data show that, akin to protein degradation, scheduled mRNA decay helps to reshape cell cycle gene expression as cells move from mitosis into G1 phase.

**\*For correspondence:**
m.tanenbaum@hubrecht.eu

[†]These authors contributed equally to this work

**Present address:** [‡]Division of Cell Biology, Oncode Institute, Netherlands Cancer Institute, Amsterdam, Netherlands

**Competing interest:** The authors declare that no competing interests exist.

## Editor's evaluation

This work is a very significant contribution on how mRNA stability regulation takes place across the cell cycle, particularly after mitosis. It also describes an original method for time-resolved transcriptome analysis during the cell cycle, which is potentially very interesting for the whole molecular and cell biology community.

## Introduction

Cell division is essential for the development and homeostasis of multicellular organisms. Precise control over cell division is paramount, as errors may contribute to carcinogenesis (*Hanahan and Weinberg, 2011*; *Malumbres and Barbacid, 2001*). In order to divide, cells pass through a number of different phases, collectively referred to as the cell cycle. The cell cycle in somatic cells consists of four phases: (1) in G1 phase a cell grows and prepares for DNA replication; (2) in S phase the DNA is replicated; (3) in G2 phase a cell prepares for segregation of the replicated genome; (4) in M phase (or mitosis) the cell divides and can then either enter into G1 phase of the next cell cycle or it can (temporarily) exit the cell cycle and enter into G0 phase (i.e. quiescence). Progression through the cell cycle is accompanied by the periodic expression of many genes (referred to as cell cycle genes), whose protein products are likely required in a particular cell cycle phase (*Bar-Joseph et al., 2008*; *Chaudhry et al., 2002*; *Cho et al., 2001*; *Cho et al., 1998*; *Grant et al., 2013*; *Whitfield et al., 2002*). Deregulated expression of cell cycle genes can decrease the fidelity of cell division. For instance, reduced expression of G2 and M phase cell cycle genes impedes mitotic entry and affects the fidelity of chromosome segregation (*Laoukili et al., 2005*). Conversely, a failure to suppress expression of G2 and M phase genes as cells enter G1 phase can result in a shortened G1 phase and cause DNA replication errors (*García-Higuera et al., 2008*; *Park et al., 2008*; *Sigl et al., 2009*), and can even contribute to carcinogenesis (*Bortner and Rosenberg, 1997*; *Coelho et al., 2015*; *Kalin et al., 2006*;

*Kim et al., 2006*; *Vaidyanathan et al., 2016*). These examples highlight the importance of tightly controlled gene expression for proper execution of the cell cycle.

To restrict cell cycle gene expression to the correct cell cycle phase, cells need to activate, but also repress the expression of cell cycle genes as they move from one phase to the next. Scheduled protein degradation plays an important role in repression of cell cycle gene expression by ensuring that protein expression is restricted to the appropriate cell cycle phase (*Nakayama and Nakayama, 2006*; *Vodermaier, 2004*). In addition, cells prevent de novo synthesis of proteins through inhibition of transcription to further restrict protein expression to the correct cell cycle phase (*Bertoli et al., 2013*; *Sadasivam and DeCaprio, 2013*). While inhibition of transcription will eventually lower mRNA levels and thus decrease protein synthesis rates, this process is relatively slow, as it requires turnover of the existing pool of mRNAs. To circumvent this, cells can shut down translation or degrade pre-existing transcripts when transitioning from one cell cycle phase to another. Indeed, control of mRNA translation also contributes to the regulation of gene expression during the cell cycle (*Kronja and Orr-Weaver, 2011*) and several hundreds of genes are subject to translational regulation at different phases of the cell cycle (*Stumpf et al., 2013*; *Tanenbaum et al., 2015*).

Regulation of mRNA stability during the cell cycle has been studied relatively little, but recent work suggests that this type of regulation also contributes to restriction of cell cycle gene expression; dynamic changes in mRNA stability during the cell cycle were observed in yeast using fluorescent in situ hybridization (FISH). Specifically, CLB2 and SWI5 mRNAs were shown to be degraded during mitosis (*Trcek et al., 2011*). Globally, mRNA synthesis and decay rates during the cell cycle of yeast were derived through metabolic mRNA labeling in synchronized populations, resulting in the identification of several hundred genes that show periodic changes in mRNA synthesis and degradation rates (*Eser et al., 2014*). Regulation of mRNA stability is also reported to occur during the human cell cycle. For instance, the transcription factor ERG was shown to control the degradation of a set of mRNAs during S phase (*Rambout et al., 2016*). Recently, global mRNA synthesis and degradation rates during the human cell cycle were determined (*Battich et al., 2020*). In this study, a newly developed method using a pulse-chase labeling approach simultaneously quantifies metabolically labeled and pre-existing unlabeled transcripts in individual cells. This method was used to determine synthesis and degradation rates of individual transcripts during the cell cycle. Together, these studies demonstrate that the stability of many mRNAs change during the cell cycle. However, due to the relatively long measurement time required for pulse-chase approaches (up to 6 hr), accurate dynamics and rapid changes, especially around the transition points in the cell cycle, are difficult to determine.

To obtain a highly quantitative view of transcriptome dynamics during cell cycle phase transitions, we established a method that combines singe-cell mRNA sequencing and live-cell imaging of cell cycle progression to map transcriptome-wide mRNA expression levels with high temporal resolution during the cell cycle. We focus specifically on the mitosis-to-G1 (M-G1) phase transition, during which cells divide and enter into a new cell cycle. This cell cycle phase transition was selected as gene expression needs to be 'reset' after cell division, requiring major changes to gene expression. The widespread protein degradation that occurs during the M-G1 phase transition is thought to contribute to this reset (*Castro et al., 2005*; *Harper et al., 2002*; *Peters, 2002*; *Vodermaier, 2004*). We hypothesized that, analogous to scheduled protein degradation, mRNA decay might play an important role in resetting cell cycle gene expression by limiting the carry-over of pre-existing G2/M-specific transcripts from one cell cycle into the next. Using our method, we identified two temporally distinct waves of mRNA decay: the first wave is initiated during mitosis and the second wave is initiated within the first hours of G1 phase. For several of these genes, we show that mRNA decay is stimulated by CNOT1, a subunit of the CCR4-NOT mRNA deadenylase complex that shortens the poly(A) tail of mRNAs, generally resulting in their decay (*Garneau et al., 2007*; *Yamashita et al., 2005*). Together, our findings demonstrate that, analogous to protein degradation, scheduled mRNA degradation occurs at the M-G1 phase transition. Scheduled mRNA degradation likely provides an important contribution to the reset of the transcriptome after cell division.

## Results

## Time-resolved transcriptome profiling during the cell cycle using the FUCCI system

To obtain a detailed view of mRNA levels as cells progress from M phase into G1 phase, we developed a method that connects live-cell microscopy with single-cell RNA sequencing (scRNA-seq), through fluorescence activated cell sorting (FACS). This method allows us to assign an accurate, 'absolute' cell cycle time (i.e. the time in minutes since the completion of metaphase) to individual sequenced cells, which we used to generate a high-resolution, time-resolved transcriptome profile of the M-G1 phase transition.

To assign an absolute cell cycle time for each cell, we expressed the fluorescent, ubiquitination-based cell cycle indicator (FUCCI) system in a human untransformed cell line, RPE-1 (RPE-FUCCI). In the FUCCI system an orange fluorescent protein (FUCCI-G1) is expressed in G1 and early S phase cells, as well as in G0 phase (quiescence), while a green fluorescent protein (FUCCI-G2) is expressed in late S, G2, and early M phase (*Figure 1A–C*, *Figure 1—figure supplement 1A* and *Figure 1—video 1*; *Sakaue-Sawano et al., 2008*). Importantly, the expression levels of both fluorescent markers change over time even within each cell cycle phase, potentially allowing precise pinpointing of the cell cycle time of individual cells based on the fluorescence intensity of the FUCCI reporter. We used live-cell microscopy to measure FUCCI-G1 fluorescence intensity in cells as they progressed through G1 phase, which revealed a monotonic increase during the first 6–8 hr after G1 entry (*Figure 1—figure supplement 1B*). To allow accurate calculation of a cell cycle time based on FUCCI-G1 fluorescence intensity, we fit the average FUCCI-G1 fluorescence intensity to a polynomial equation (*Figure 1D* and *Supplementary file 1*), which generates an accurate mathematical description of the data. Using this equation, a cell cycle time can be calculated for each cell based on its fluorescence intensity of the FUCCI-G1 marker. Cell cycle times of individual cells can be accurately determined during the first ~5 hr of G1/G0 phase, after which prediction accuracy decreases due to the increased cell-to-cell heterogeneity in FUCCI-G1 fluorescence at later time points in G1 phase (*Figure 1—figure supplement 1B-C*).

In the scRNA-seq protocol the FUCCI fluorescence intensity of each sequenced cell is measured by FACS during FACS of cells into 384-well plates. To assess precise cell cycle times based on FUCCI fluorescence intensities measured by FACS, fluorescence intensities measured by FACS need to be compared with those obtained by microscopy. For this, we normalized FUCCI-G1 fluorescence intensities from both assays. Early S phase cells can be identified in both live-cell imaging experiments and by FACS analysis (*Figure 1C*, *Figure 1-figure supplement 1A and D*), and since the mean fluorescence intensity of the FUCCI-G1 marker in early S phase is constant, it can be used as a normalization factor to directly compare the FUCCI-G1 fluorescence intensity values obtained by imaging and FACS (*Figure 1C*, *Figure 1-figure supplement 1D*; see Materials and methods). Using this normalization factor and the fluorescence intensity of the FUCCI-G1 marker as assayed by FACS, it is possible to map individual G1 cells assayed by FACS onto time-lapse microscopy data, allowing us to pinpoint the precise cell cycle time of each cell.

To validate our method of converting FACS fluorescence intensities into absolute cell cycle times, we performed an alternative method to determine cell cycle times based on FUCCI fluorescence as measured by FACS; we blocked cells in mitosis using the microtubule stabilizing drug Taxol for various durations, preventing entry of cells in G1 phase. For cells already in G1 phase the FUCCI-G1 fluorescent signal continues to increase. As no new cells enter G1 phase, a gradual loss of cells with low FUCCI-G1 fluorescence is observed by FACS (*Figure 1—figure supplement 1E*). By mapping the population of cells that is lost after different durations of Taxol treatment we could calculate the FUCCI-G1 fluorescence intensity associated with cells that had spent various times in G1 phase. Comparison of both methods revealed very similar cell cycle times (*Figure 1—figure supplement 1F*). Thus, we conclude that we can accurately determine the time a cell has spent in G1 phase based on its FUCCI-G1 fluorescence as measured by FACS.

To identify changes to the transcriptome throughout the M-G1 phase transition, we FACS-isolated single G2, M, and G1 phase cells based on their FUCCI-G1 and FUCCI-G2 fluorescence (*Figure 1C*), and subjected them to scRNA-seq. In total, 1152 cells were sequenced in three replicate experiments, of which 841 cells passed quality checks (see Materials and methods) and were used to generate a high-resolution temporal transcriptome profile of the M-G1 phase transition. Since the FUCCI system

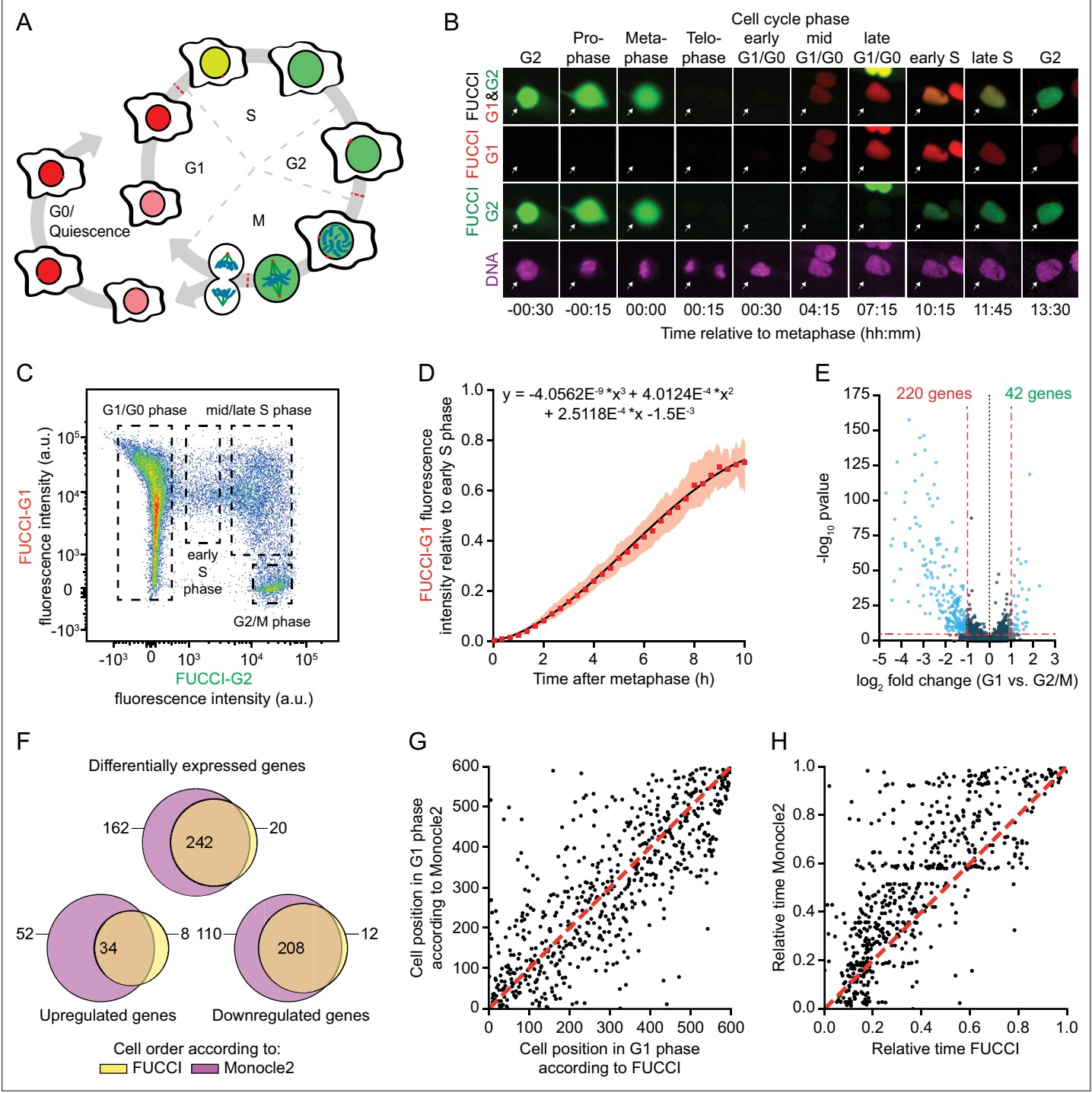

**Figure 1.** A method for time-resolved transcriptome analysis during the cell cycle. (**A**) Schematic representation of the human cell cycle and the fluorescent, ubiquitination-based cell cycle indicator (FUCCI) system. (**B**) Representative images of RPE-FUCCI cells throughout the cell cycle (see *Figure 1—video 1*). RPE-FUCCI cells were incubated for 2 hr with SPY650-DNA to visualize the DNA. Cells were imaged every 15 min for a duration of 15 hr. Arrows indicate a single cell undergoing a complete cell cycle. (**C**) Fluorescence activated cell sorting (FACS) analysis of asynchronously growing RPE-FUCCI cells. Dashed boxes indicate the gating strategy used for the identification and isolation of cells in G1/G0, early S phase, mid/late S phase, and G2/M phases. (**D**) Modeling of FUCCI-G1 fluorescence intensities. Asynchronously growing RPE-FUCCI cells were analyzed by live-cell imaging (*Figure 1—figure supplement 1A*). Subsequently, FUCCI-G1 fluorescence intensities were measured and normalized to the average FUCCI-G1 fluorescence in early S phase cells (see Materials and methods). Red squares and shading represent the mean fluorescence and SEM, respectively, of three individual experiments. The mean FUCCI-G1 fluorescence was fit to a third-order polynomial (black line, equation above plot). The fit has no biological meaning, but serves to approximate the data to allow calculation of G1 phase cell cycle times based on FUCCI-G1 fluorescence intensity.

*Figure 1 continued on next page*

**eLife** Research article

Cell Biology | Chromosomes and Gene Expression

*Figure 1 continued*

(**E**) Differential gene expression analysis of RPE-FUCCI cells in G2/M phase versus G1 phase. Cells were ordered based on FUCCI fluorescence and differential gene expression analysis was performed using Monocle2 (see Materials and methods). (**F**) Venn diagram comparing differentially expressed genes (both up- and downregulated in G1 versus G2/M phase) identified after FUCCI- or Monocle2-based cell ordering. (**G**) Comparison of FUCCI- and Monocle2-based ordering of G1 phase cells. Dashed red line indicates identical order of cells. (**H**) Comparison of G1 phase cell cycle time from FUCCI-based ordering with G1 pseudo time based on trajectory inference by Monocle2. Both FUCCI-G1 phase cell cycle time and Monocle2 G1 pseudo time are normalized to values between 0 and 1 for comparison. Dashed line indicates identical timing of FUCCI and Monocle2.

The online version of this article includes the following video and figure supplement(s) for figure 1:

**Figure supplement 1.** A method for time-resolved transcriptome analysis during the cell cycle.

**Figure 1—video 1.** Example movie of RPE-1 cells expressing the G1 and G2 fluorescent, ubiquitination-based cell cycle indicator (FUCCI) reporters. https://elifesciences.org/articles/71356/figures#fig1video1

does not discriminate between cells in G2 and M phase, and as there are few transcriptome changes between these two phases (*Tanenbaum et al., 2015*), we averaged the transcript levels of all cells in G2 and M phase (referred to as G2/M). The average G2/M expression levels of individual genes displayed a high correlation between different replicate experiments ($\rho$ = 0.94–0.95) (*Figure 1—figure supplement 1G-I*), allowing us to pool the data from the different experiments. The final dataset consisted of 86 G2/M phase cells and 755 cells from various time points in G1 phase (up to 9 hr after the M-G1 phase transition) (*Figure 1—figure supplement 1J* and *Supplementary file 1*). After initial data processing, we performed differential transcriptome analysis comparing G2/M phase to G1 phase (see Materials and methods). This analysis identified 220 genes that were downregulated and 42 genes that were upregulated when cells from G2/M phase were compared with G1 phase cells (using a cutoff of >2-fold expression change, see Materials and methods) (*Figure 1E* and *Supplementary file 1*). Gene Ontology analysis revealed that these differentially expressed genes were strongly enriched for cell cycle functions, as expected (*Figure 1—figure supplement 1K*). Of all genes involved in the cell cycle (derived from Cyclebase 3.0; *Santos et al., 2015*), for which we could determine the expression, ~53% is downregulated in early G1 phase in our dataset (>2-fold) (*Figure 1—figure supplement 1L* and *Supplementary file 1*).

To compare our method of cell cycle time determination with previous computational methods of (pseudo) time determination, we used Monocle2, an in silico trajectory inference method that orders cells based on their transcriptomes (*Qiu et al., 2017a*; *Qiu et al., 2017b*; *Trapnell et al., 2014*). We aligned cells using trajectory inference (*Figure 1—figure supplement 1M*, see Materials and methods), and subsequently performed differential transcriptome analysis, which identified 318 downregulated genes and 86 upregulated genes in early G1 phase compared to G2/M phase (*Figure 1—figure supplement 1N* and *Supplementary file 1*). There was a large overlap between the differentially expressed genes identified by Monocle2 and our FUCCI-based method (*Figure 1F*), and we found a good overall correlation between FUCCI-based ordering and Monocle2-based ordering of G1 phase cells (*Figure 1G* and *Figure 1—figure supplement 1M*). Monocle2 cannot assign absolute cell cycle times, instead it can compute a 'pseudo time' for each G1 phase cell assuming that transcriptome changes occur evenly over time. Comparing the pseudo time assigned by Monocle2 with the cell cycle time assigned by our FUCCI-based method revealed differences between both methods. In general, Monocle2 computed larger time intervals between cells early in G1 phase compared to our FUCCI-based method (*Figure 1H*). As Monocle2 computes the time intervals between cells based on the magnitude of transcriptome changes, a possible explanation for this observation is that transcriptome changes are larger in early G1 phase than at the end of G1 phase, and Monocle2 thus positions cells in early G1 phase too far apart in (pseudo) time. In conclusion, by using the FUCCI-based single-cell sequencing approach we could generate a high-resolution, time-resolved transcriptome profile of cells spanning the transition from M phase into G1 phase.

## mRNA levels decline in multiple waves during the M-G1 phase transition

As discussed above, we found a large group of genes (220) for which mRNA levels decline at the M-G1 phase transition. To determine the precise moment when mRNA levels started to decline for each gene, we fit the data for individual mRNAs to a smoothing spline and determined the moment

of maximum negative slope of the spline, which is the moment when the mRNA level declined most rapidly (referred to as the 'spline analysis'; see Materials and methods). Strikingly, the decline in mRNA levels of various genes initiated at two distinct times in the cell cycle. The first 'wave' of mRNA decline occurred within 20 min of metaphase (*Figure 2A* and *Supplementary file 1*), which is around the time of mitotic exit, but before the start of G1 phase (*Figure 1B*). The second wave occurred at ~80 min after metaphase (*Figure 2A* and *Supplementary file 1*), which is in G1 phase (G1 phase starts between 15 and 30 min after metaphase; *Figure 1B*). To examine these two 'waves' of mRNA decline in more detail, we divided the 220 mRNAs into two groups: for the first group the maximum negative slope occurred during mitotic exit (*immediate decrease*) and for the second group the maximum negative slope occurred during early G1 phase (*delayed decrease*) (*Supplementary file 1*, see Materials and methods). Plotting the average slope over time for both groups revealed that the mRNAs in the *immediate decrease* group declined most rapidly during the M-G1 phase transition and continued to decline during the first 2–3 hr of G1 phase (*Figure 2B*, red lines), whereas the mRNAs in the *delayed decrease* group mostly declined between 1 and 4 hr after the start of G1 phase (*Figure 2B*, blue lines). For both groups, the slopes of individual mRNAs were mostly centered around zero at later times (>8 hr) in G1 phase, demonstrating that most mRNAs reached a new steady-state level at later time points in G1 phase.

To confirm that mRNA levels decline in two distinct temporal waves, cells were isolated by FACS (*Figure 2—figure supplement 1A*) and RT-qPCR was used to measure mRNA levels for five genes in the *immediate decrease* group (CDK1, TOP2A, UBE2C, FBXO5, and FZR1) and five genes in the *delayed decrease* group (ARL6IP1, CENPA, PSD3, UBALD2, and SRGAP1) in G2/M phase and at various time points in G1 phase. Consistent with the RNA sequencing data, we observed two distinct waves of mRNA decline by RT-qPCR (*Figure 2C*). Note that the minor increase in mRNA levels seen at the 1 hr time point in the *delayed decrease* group is likely an artifact caused by comparing a highly synchronized population of early G1 phase cells (1 hr time point, when cells have not yet initiated the decline of delayed genes and thus express the highest possible levels of these transcripts) to a somewhat more heterogeneous population of G2/M phase cells (0 hr time point).

To determine the moment of mRNA decline relative to cell division more precisely for the *immediate decrease* group, we assessed mRNA levels by single molecule fluorescence in situ hybridization (smFISH) and fluorescence microscopy during different stages of mitosis. We fixed asynchronously growing cultures of cells and stained them for two mRNAs from the *immediate decrease* group, TOP2A and CDK1, which were selected because of strong mRNA decline after metaphase (*Figure 2C*). To determine the mitotic stages and the outline of the individual cells, we stained the DNA with DAPI, and the membranes with fluorescent wheat germ agglutinin (WGA) (*Figure 2D*). Quantification of TOP2A and CDK1 mRNA levels at various stages of mitosis revealed a significant decrease in mRNA levels as early as anaphase, and a further decrease in telophase for both genes (*Figure 2E–F* and *Figure 2—figure supplement 1B-C*).

To further confirm the moment of mRNA decline of mRNAs belonging to the *immediate decrease* group, we assessed the moment of mRNA decline for additional mRNAs belonging to this group by RT-qPCR. We compared mRNA levels in G2 phase to mRNA levels in early mitosis (prometaphase/metaphase) and late mitosis (anaphase/telophase). G2 phase cells as well as cells in early and late mitosis were isolated by a combination of mitotic shake-off and FACS (*Figure 2—figure supplement 1D-F*, see Materials and methods). RT-qPCR analysis revealed that the levels of all five genes belonging to the *immediate decrease* group decline as cells move beyond metaphase (*Figure 2—figure supplement 1G*). In contrast, the levels of all five transcripts belonging to the *delayed decrease* group do not decline between G2 phase, early and late mitosis (*Figure 2—figure supplement 1H*). FBXO5 and UBE2C mRNA levels already show decline as cells move from G2 phase into early mitosis, while the levels of CDK1, TOP2A, and FZR1 mRNAs do not (their levels increase slightly between G2 phase and mitosis, which may reflect a cell synchronization artifact, as discussed before) (*Figure 2—figure supplement 1G*). These results suggest that genes belonging to the *immediate decrease* group decrease already in mitosis, while genes belonging to the *delayed decrease* group do not start to decrease until the onset of G1 phase.

Our previous analysis revealed two waves of mRNA decline. Since these waves of mRNA decline are determined based on cell populations, it is unclear if both waves occur in each individual cell, or whether these two waves rather occur in two different sub-populations of cells, for example in cells

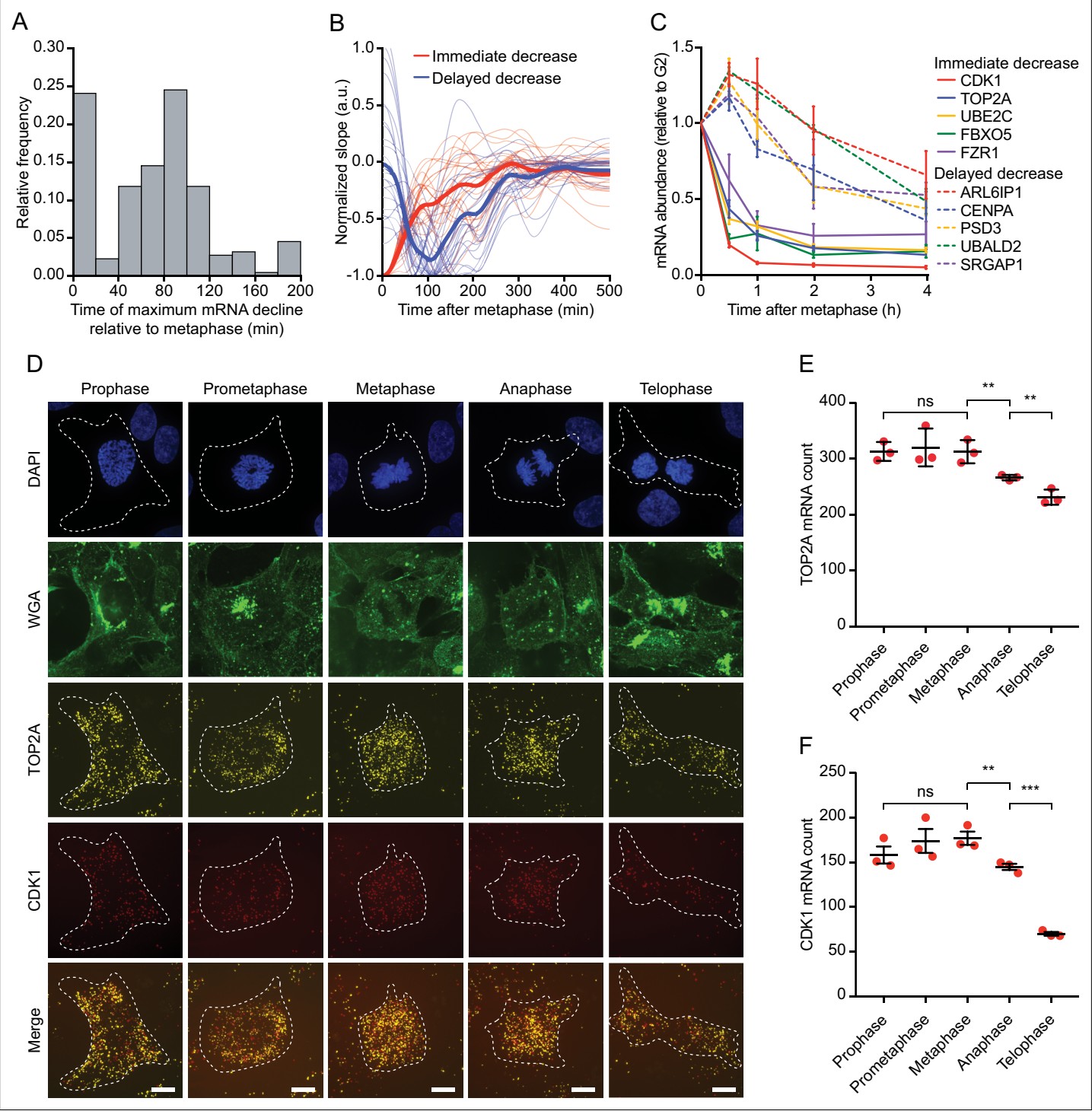

**Figure 2.** Reduction in mRNA levels occurs in multiple waves, during and after cell division. (**A**) Time relative to metaphase of the highest rate of mRNA decrease for the 220 downregulated genes. For each gene a smoothing spline was fit to the data and the moment of maximum negative slope of the spline was determined (see Materials and methods). (**B**) Average slope of mRNA levels over time for genes that display *immediate* (thick red line) or *delayed decrease* (thick blue line). Thin red and blue lines show a random selection of 25 individual genes belonging to the *immediate* or *delayed decrease* group, respectively. (**C**) Validation of the two waves of mRNA decline. RPE-fluorescent, ubiquitination-based cell cycle indicator (FUCCI) cells at different stages of the cell cycle were isolated by fluorescence activated cell sorting (FACS) based on FUCCI fluorescence (see *Figure 2—figure supplement 1A* for gating strategy). mRNA expression levels of indicated genes was measured by RT-qPCR. Five genes from the *immediate decrease* group and five genes from the *delayed decrease* group were selected. Note that the moment of decrease as measured by RT-qPCR closely mirrors the moment of decrease determined by modeling of our single-cell sequencing data (see *Supplementary file 1*). Lines with error bars represent average

*Figure 2 continued on next page*

*Figure 2 continued*

± SEM of three experiments. (**D**) Example images of TOP2A and CDK1 single molecule fluorescence in situ hybridization (smFISH) at the different stages of mitosis. Asynchronously growing RPE-1 cells were fixed and stained for DNA (DAPI), membranes (WGA), and TOP2A and CDK1 mRNA (using smFISH). Scale bar, 10 μm. (**E–F**) Quantification of TOP2A (**E**) and CDK1 (**F**) transcript number in different stages of mitosis. Each dot represents the average number of transcripts in a single experiment and lines with error bars represent average ± SEM of three experiments (at least 15 cells per experiment per condition analyzed, see *Supplementary file 2* for the exact number of cells included). Single-cell TOP2A and CDK1 transcript counts are shown in *Figure 2—figure supplement 1B-C*. p-Values are based on a one-tailed unpaired Student's t-test, and are indicated as * (p < 0.05), ** (p < 0.01), *** (p < 0.001), ns = not significant.

The online version of this article includes the following figure supplement(s) for figure 2:

**Figure supplement 1.** Reduction in mRNA levels occurs in multiple waves, during and after cell division.

entering G1 or G0 phase (which cannot be distinguished based on the FUCCI-G1 reporter). To assess whether both waves of mRNA decline occur in cells entering G1 (rather than G0), we generated p53 knock-out cells (*Figure 2—figure supplement 1I-J*), which exclusively enter G1 after completion of mitosis (*Figure 2—figure supplement 1K*; *Spencer et al., 2013*; *Yang et al., 2017*). We confirmed that the synthesis rates of the FUCCI-G1 fluorescent reporter are unaltered in RPE-FUCCI Δp53 cells (*Figure 2—figure supplement 1L*), and assessed whether two waves of mRNA decrease occurred in G1 phase by RT-qPCR (see *Figure 2—figure supplement 1A*). We found that mRNA levels declined in two distinct waves (compare *Figure 2—figure supplement 1M* and *Figure 2C*), demonstrating that both waves of mRNA decline occur in G1 phase cells.

To further confirm that the two waves of mRNA decline occur in individual cells, we examined the expression of *immediate decrease* and *delayed decrease* genes in the same cells. If the two waves of mRNA decline occur in a distinct subset of cells, then cells showing the strongest decrease in the one set of genes (e.g. *immediate decrease* genes) should show little to no decrease in the other set of genes (e.g. *delayed decrease* genes). In contrast, if both waves of mRNA decline occur sequentially in the same cells, such anti-correlation is not expected. To investigate this, we averaged the expression of all *immediate decrease* and *delayed decrease* genes and compared the expression of both groups of genes in single cells. A time point of 4 hr after metaphase was selected for this analysis, as both types of mRNA decline (*immediate* and *delayed*) have been largely competed at this time point. We found that the expression levels of both groups of genes are highly correlated in individual cells (*Figure 2—figure supplement 1N*), demonstrating that both waves of mRNA decline occur in the same cells.

## mRNA decay drives transcriptomic changes during the M-G1 phase transition

The decline in mRNA levels during early G1 phase may be caused by changes in the rate of mRNA synthesis (transcription) and/or degradation (mRNA stability). It is well established that transcription regulation controls the expression of cell cycle genes (*Bertoli et al., 2013*; *Sadasivam and DeCaprio, 2013*). Accordingly, comparison of transcription rates in G2 phase and G1 phase RPE-1 cells (data derived from *Battich et al., 2020*) shows that transcription is decreased for nearly all genes that are downregulated in G1 phase compared to G2/M phase (*Figure 3A*, see Materials and methods), regardless of whether mRNAs belong to the *immediate decrease* or *delayed decrease* group (*Figure 3—figure supplement 1A*). We wondered if transcription inhibition alone was sufficient to explain the rapid rate at which transcript levels decline in G1 phase, or whether an increase in mRNA degradation also contributes to the decreased expression in G1 phase for the genes that we find to be downregulated in G1 phase. To investigate whether the mRNA degradation rate is altered during the M-G1 phase transition, we calculated the degradation rate of individual mRNAs during the M-G1 phase transition using mathematical modeling (*Figure 3B*, see Materials and methods). Briefly, our model describes two phases for the mRNA levels over time: in the first phase, mRNA levels remain constant (at an initial level of $m_0$), in the second phase mRNA levels decline to a new steady-state level. The onset of decline is described by $t_{onset}$. The rate of decline is dependent on the mRNA degradation rate ($\gamma$), while the new steady-state mRNA level is dependent on the mRNA synthesis rate ($\mu$) and on the mRNA degradation rate ($\gamma$). Using a quality of fit analysis (see Materials and methods), we identified the parameters ($m_0$, $t_{onset}$, $\mu$, and $\gamma$) that resulted in the optimal fit with the data for each of the 220 downregulated genes. Visual inspection showed that the fits described the data well

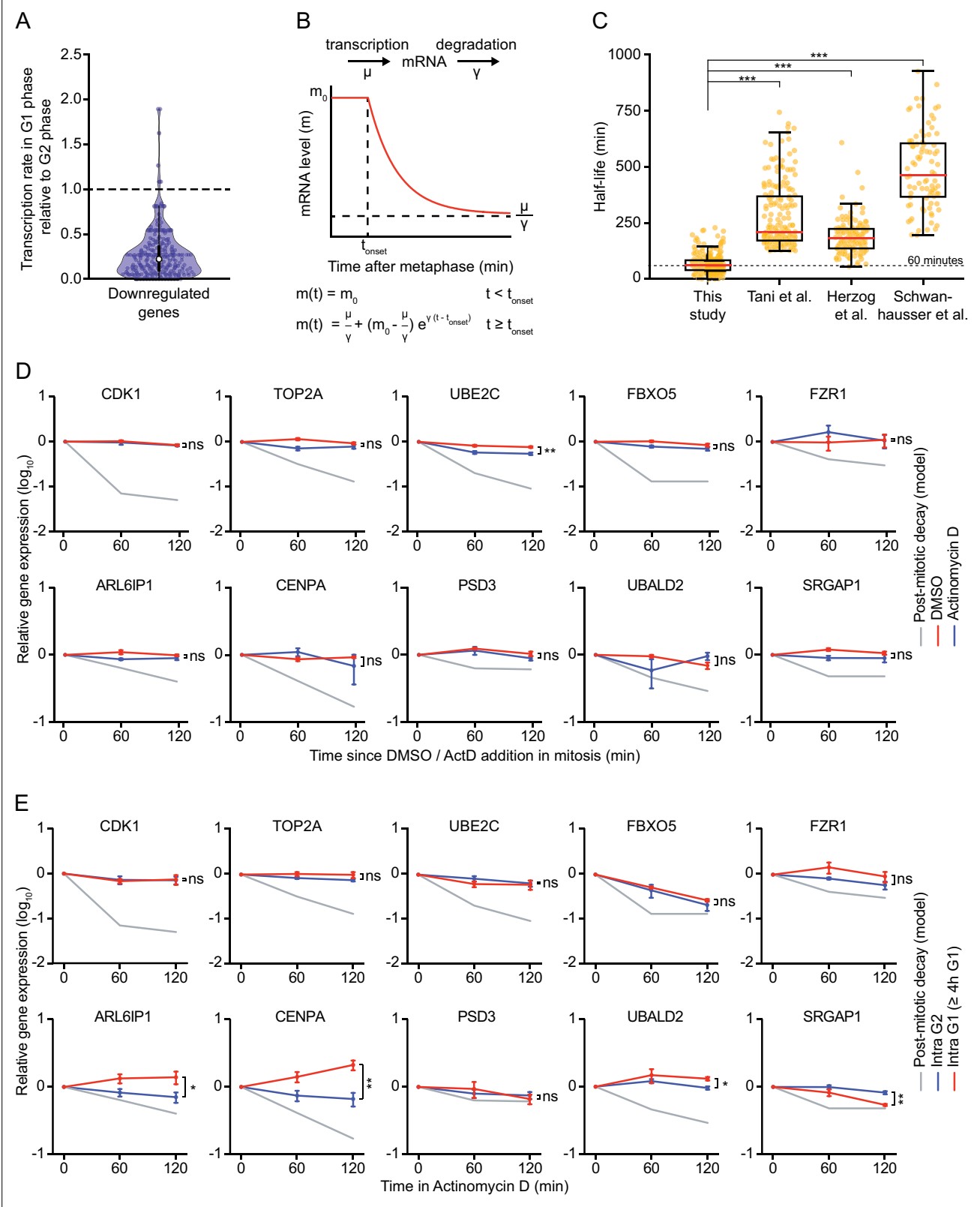

**Figure 3.** mRNA decay occurs during a brief window of time as cells exit mitosis and enter G1 phase. (**A**) Violin plot showing the ratio of transcription in G1 phase versus G2 phase for the 220 genes that we identified as downregulated in G1 phase. Data were retrieved from *Battich et al., 2020*. Battich et al. labeled new transcripts for 30 min using EU, and old and new transcripts were quantified using deep sequencing. We defined the relative rate of transcription as the number of labeled transcripts in G1 versus G2 phase. Dashed line indicates a ratio of 1, indicative of a similar transcription rate in G2

*Figure 3 continued*

and G1 phase (a ratio <1 is indicative of reduced transcription in G1 phase). (**B**) Schematic of the mathematical model that was used to fit the decrease in mRNA levels as cells progress from mitosis into G1 phase. (**C**) Boxplot of mRNA half-lives for the genes that were identified as downregulated in G1 phase in our study. Half-lives at the mitosis-to-G1 (M-G1) transition are shown (this study), as well as the half-lives of the same genes determined in asynchronous cell populations in HeLa cells (Tani et al.), mouse embryonic stem cells (Herzog et al.) and mouse fibroblasts (Schwanhausser et al.). (**D**) Relative mRNA levels in mitosis after different times of transcription inhibition, as measured by RT-qPCR. RPE-1 cells were synchronized in G2 using the CDK1-inhibitor RO-3306. Subsequently, cells were released from RO-3306 into medium containing Taxol, to arrest cells in mitosis. Mitotic cells were collected by mitotic shake-off, and cultured for an additional 2 hr in the presence or absence of the transcription inhibitor Actinomycin D (blue and red lines, respectively). For comparison, mRNA levels during the M-G1 phase transition are shown (gray line). Note that mRNA of indicated genes is stable in mitosis, indicating that mRNA is degraded specifically during the M-G1 phase transition. Lines with error bars indicate average ± SEM of three experiments. (**E**) Relative mRNA levels in G2 and late G1 phase after different times of transcription inhibition, as measured by RT-qPCR. Asynchronously growing RPE-fluorescent, ubiquitination-based cell cycle indicator (FUCCI) cells were treated with Actinomycin D for indicated times. Cells were then fluorescence activated cell sorting (FACS)-sorted and G2 phase cells and late G1 phase cells (>4 hr into G1 phase) were isolated based on FUCCI reporter fluorescence. The mRNA levels of indicated genes were then measured by RT-qPCR. mRNA levels during the M-G1 phase transition are shown for comparison (gray lines). Note that mRNA levels are substantially less stable in cells during the M-G1 phase transition compared to G2 or late G1 phase cells. Lines with error bars indicate average ± SEM of three experiments. p-Values are based on a one-tailed unpaired Student's t-test (**C-E**), and are indicated as * (p < 0.05), ** (p < 0.01), *** (p < 0.001), ns = not significant.

The online version of this article includes the following figure supplement(s) for figure 3:

**Figure supplement 1.** mRNA decay occurs during a brief window of time as cells exit mitosis and enter G1 phase.

(*Figure 3—figure supplement 1* and *Supplementary file 1*). Using this approach, we confirmed that the onset of decay for different genes occurred most strongly at two distinct times during the M-G1 phase transition; either during the M-G1 phase transition or during early G1 phase (*Figure 3—figure supplement 1H*), confirming the results from the spline analysis (*Figure 2A*).

We used mRNA degradation rates extracted from the model to compute the half-lives of the 220 mRNAs that we found to be downregulated in G1 phase (*Supplementary file 1*). This revealed a median half-life of 61.5 min in the decay phase during the M-G1 phase transition (*Figure 3—figure supplement 1I*). We compared mRNA half-lives during the M-G1 phase transition with previously determined half-lives of the same mRNAs in asynchronously growing cells. For almost all mRNAs (98–100%) the M-G1 half-lives we computed are shorter than the reported half-lives of the same mRNAs in asynchronously growing cells (*Figure 3C* and *Figure 3—figure supplement 1J-L*; *Herzog et al., 2017*; *Schwanhäusser et al., 2011*; *Tani et al., 2012*). We observed no significant differences between the half-lives of mRNAs belonging to the *immediate decrease* group versus the *delayed decrease* group (*Figure 3—figure supplement 1M*). The comparatively short mRNA half-lives we find during the M-G1 phase transition indicate that these transcripts are subject to scheduled degradation at this stage of the cell cycle.

To confirm that mRNAs are subjected to scheduled degradation specifically during the M-G1 phase transition, we also examined their stability during mitosis, G2 phase and late G1 phase. To measure mRNA stability in mitosis, we synchronized and arrested RPE-1 cells in prometaphase of mitosis (see Materials and methods), followed by inhibition of transcription for 1 or 2 hr using Actinomycin D. Actinomycin D completely blocked de novo transcription (*Figure 3—figure supplement 1N*) and did not influence the arrest of cells in mitosis (*Figure 3—figure supplement 1O*). None of the 10 mRNAs tested (belonging to both the *immediate* and *delayed decrease* groups) showed an appreciable decrease in mRNA levels during the 2 hr time window of Actinomycin D treatment, indicating that these mRNAs are much more stable in mitosis than they are during the M-G1 phase transition (*Figure 3D*, compare red or blue lines to gray line).

To measure mRNA stabilities in G2 phase and late G1 phase, we inhibited transcription with Actinomycin D for 1 or 2 hr in asynchronously growing RPE-FUCCI cells. Subsequently, we FACS-sorted populations of G2 phase cells and late G1 phase cells (cells that had spent at least 4 hr in G1 phase) and determined mRNA levels of immediate and delayed decay genes with or without Actinomycin D treatment. For all genes tested, mRNA stability in G2 phase and late G1 phase substantially exceeded the mRNA stability calculated during the M-G1 phase transition (*Figure 3E*, compare red or blue lines to gray line). Collectively, these data demonstrate that for all genes tested, mRNAs are substantially more stable during G2 phase, mitosis (pre-anaphase), and late G1 phase compared to during the M-G1 phase transition and early G1 phase. Thus, these results uncover an active mRNA decay mechanism that specifically takes place during mitotic exit and early G1 phase.

## CNOT1 stimulates mRNA decay during the M-G1 phase transition

Cytoplasmic mRNA degradation is often initiated by shortening of the poly(A) tail (*Eisen et al., 2020*), followed by degradation from either end of the mRNA (*Garneau et al., 2007*). Shortening of the poly(A) tail is frequently mediated by the CCR4-NOT complex (*Yamashita et al., 2005*). To test whether the CCR4-NOT complex is required for mRNA decay during the M-G1 phase transition, we depleted CNOT1, the scaffold subunit of the CCR4-NOT complex, using siRNA-mediated depletion in RPE-1 cells (*Figure 4—figure supplement 1A*). For initial experiments, we focused on TOP2A and CDK1 mRNAs, both of which are rapidly and robustly degraded at the M-G1 phase transition (*Figure 2C*). CNOT1 depletion resulted in substantially fewer mitotic cells and a strong enrichment of cells in G1 phase (*Figure 4—figure supplement 1B*), consistent with an important role for CNOT1 in cellular proliferation (*Blomen et al., 2015*; *Hart et al., 2015*; *Wang et al., 2015*). In the mitotic cells that could be identified, depletion of CNOT1 caused a 20–25% increase in the relative abundance of both TOP2A and CDK1 mRNAs in telophase (when decay of these mRNAs has normally occurred) compared to control cells (*Figure 4A* and *Figure 4—figure supplement 1C-D*). These results suggest that CNOT1-dependent mRNA deadenylation is involved in mRNA decay at the M-G1 phase transition.

A previous study found that the mRNAs of many cell cycle genes contain significantly shorter poly(A) tails in M phase compared to S phase (*Park et al., 2016*). Interestingly, re-analysis of this previous data revealed that the poly(A) tails of transcripts in the *immediate decrease* group were shorter than those of genes that did not show mRNA decay at the M-G1 phase transition (median value 58 versus 79, respectively) (*Figure 4B*). The *delayed decrease* group mRNAs also have significantly shorter poly(A) tails during mitosis than those in the control group, although the effect was modest (median value 75 versus 79, respectively) (*Figure 4B*). Importantly, the shortened poly(A) tails were specific to mitosis, as poly(A) tail lengths in S phase of *immediate* and *delayed decrease* groups were similar to those of control genes (*Figure 4C*). Collectively, these data show that CNOT1 aids the decay of TOP2A and CDK1 mRNAs during the M-G1 phase transition and suggest that CNOT1-dependent deadenylation in mitosis may contribute to the decay of many mRNAs at the M-G1 phase transition.

To determine whether CNOT1 is also involved in the second wave of mRNA decay, we depleted RPE-FUCCI cells of CNOT1 using CRISPR interference (CRISPRi) (*Gilbert et al., 2014*; *Gilbert et al., 2013*). Using CRISPRi, we could knock down CNOT1 in a large population of cells, allowing subsequent FACS-based isolation of sufficient numbers of early G1 phase cells. We used three independent single-guide RNAs (sgRNAs) to target CNOT1 by CRISPRi, which resulted in a modest (~50%) reduction of CNOT1 mRNA levels (*Figure 4—figure supplement 1E*). Nonetheless, the modest reduction of CNOT1 mRNA levels still caused a cell cycle arrest (*Figure 4—figure supplement 1F*), albeit to a lesser extent than the arrest caused by the more efficient siRNA-mediated depletion of CNOT1 (compare *Figure 4—figure supplement 1B and F*). CNOT1 depletion did not affect the synthesis rates of the FUCCI-G1 fluorescent reporter (i.e. accumulation of fluorescence over time) during G1 phase (*Figure 4—figure supplement 1G*), allowing us to isolate control and CNOT1-depleted cells at similar times in G1 phase based on FUCCI fluorescence through FACS, as before (see *Figure 2—figure supplement 1A*). G2/M phase and early G1 phase cell populations were isolated by FACS and mRNAs of the *immediate decrease* and *delayed decrease* groups were measured by RT-qPCR. Even though depletion of CNOT1 was modest in these experiments, a small but reproducible decrease in decay was observed for eight of the ten genes tested (*Figure 4D*). Taken together, these data demonstrate that both waves of post-mitotic mRNA decay are stimulated by CNOT1.

## Discussion

### Assigning a precise cell cycle time to individual, sequenced cells

Understanding of gene expression control has flourished due to the development of single-cell sequencing techniques. To investigate transcriptome changes over time, trajectory inference methods have been developed that allow in silico ordering of cells, based on (dis)similarities in transcriptomes (*Saelens et al., 2019*). This creates single-cell trajectories of a biological process of interest, such as differentiation or the cell cycle (*Figure 1—figure supplement 1M*; *Haghverdi et al., 2016*; *Trapnell et al., 2014*), and is useful to study dynamics in gene expression. However, due to clustering based on transcriptome (dis)similarities, pseudo time may under/overestimate true cellular state

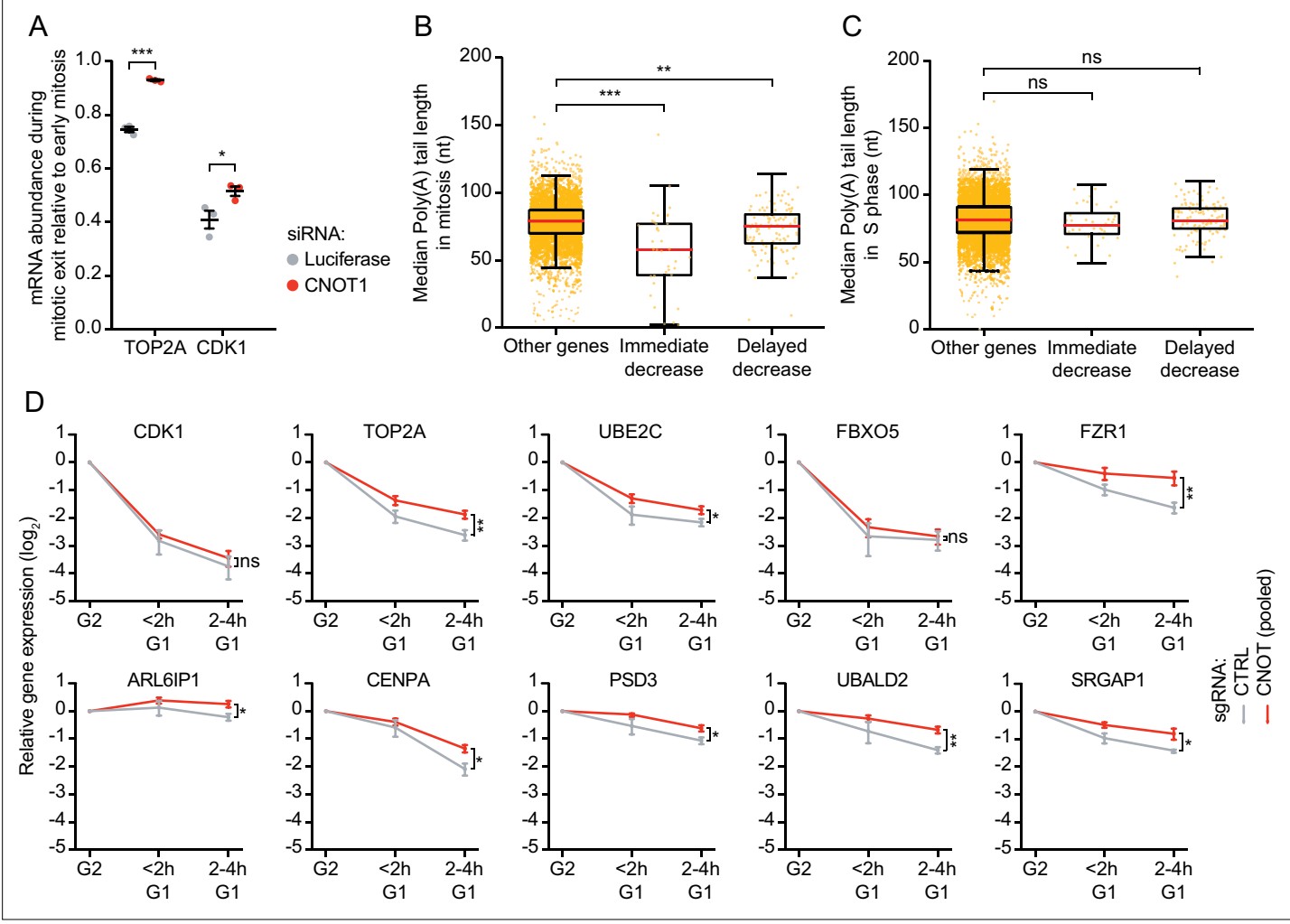

**Figure 4.** CNOT1 promotes decay of mRNAs during mitotic exit and early G1 phase. (**A**) Effect of CNOT1 depletion on TOP2A and CDK1 mRNA abundance in different states of mitosis. Cells were transfected with indicated siRNAs. Two days after transfection, cells were fixed and TOP2A and CDK1 mRNAs were visualized using single molecule fluorescence in situ hybridization (smFISH) (as in *Figure 2D*) and the number of mRNAs per cell was determined. To calculate the relative abundance of mRNAs during mitotic exit, we divided the average number of mRNAs present in telophase by the average number of mRNAs present in prophase, prometaphase, and metaphase mRNA abundance is similar during these phases of mitosis (*Figure 2E–F*). Relative abundance was used instead of absolute abundance, as the absolute number of detectable foci varied between experiments due to variations in labeling intensity of smFISH probes. Each dot represents a single experiment and lines with error bars indicate average ± SEM of three experiments. Per experiment, at least 10 cells during mitotic exit and 10 early mitotic cells were quantified (see *Supplementary file 2* for the exact number of cells included). p-Values are based on a one-tailed Student's t-test, and indicated as * ($p < 0.05$), ** ($p < 0.01$), *** ($p < 0.001$), ns = not significant. (**B–C**) Boxplot of poly(A) tail lengths in mitosis (**B**) and S phase (**C**) for *immediate decrease* genes, *delayed decrease* genes, or genes that are not subjected to mRNA decay (other genes). p-Values are based on a one-tailed Student's t-test. (**E**) Expression levels of indicated mRNAs during the mitosis-to-G1 (M-G1) phase transition in control cells and cells depleted of CNOT1. RPE-fluorescent, ubiquitination-based cell cycle indicator (FUCCI) CRISPR interference (CRISPRi) cells infected with control- or one of three different CNOT1-targeting sgRNAs were sorted into populations of G2/M phase and G1 phase cells at 5 days post-sgRNA infection. mRNA levels of indicated genes were measured by RT-qPCR. Data from three CNOT1-targeting gRNAs were averaged. Lines and error bars indicate average ± SEM of three experiments. p-Values are based on a one-tailed Welch's t-test. p-Values are indicated as * ($p < 0.05$), ** ($p < 0.01$), *** ($p < 0.001$), ns = not significant.

The online version of this article includes the following figure supplement(s) for figure 4:

**Figure supplement 1.** CNOT1 promotes decay of mRNAs during mitotic exit and early G1 phase.

durations (*Tian et al., 2019*). In addition, trajectories lack real temporal information and are therefore not ideal to determine absolute mRNA synthesis and degradation rates. To circumvent these issues, we have developed a method that combines live-cell microscopy with scRNA-seq to generate a high-resolution, time-resolved transcriptome profile in human cells. Using the FUCCI system, we

have applied our method to map gene expression dynamics during the M-G1 phase transition of the cell cycle. Even though the FUCCI system has previously been used to order single-cell transcriptomes along the cell cycle (*Battich et al., 2020*; *Hsiao et al., 2020*; *Mahdessian et al., 2021*), a unique feature of our method is that it uses precisely calibrated FUCCI reporter fluorescence intensities for accurate assignment of cell cycle times of individual, sequenced cells. We use these calibrated fluorescence intensities to align cells along the cell cycle according to their cell cycle 'age'. While cell-to-cell heterogeneity in the FUCCI fluorescence introduces some noise in the cell cycle timing of individual cells (*Figure 1—figure supplement 1B-C* and *Figure 1—video 1*), measurements on many single cells allows averaging of the timing of individual cells, resulting in a high-resolution, time-resolved transcriptome profile. Using our transcriptome profile of the M-G1 phase transition, we identify hundreds of mRNAs that show sharp transitions in their expression levels as cells progress from mitosis to G1 phase (*Figure 1E*). The availability of temporal information allowed us to quantitatively determine mRNA degradation rates for these transcripts (*Figure 3B–C* and *Supplementary file 1*), which is not possible using trajectory inference methods. Importantly, our quantitative measurements of mRNA stability allowed us to differentiate mRNA decline at the M-G1 phase transition by combined transcription inhibition and mRNA degradation from transcription inhibition alone. Similar approaches are likely possible for other biological events for which fluorescence reporters are available, making this approach a broadly applicable method.

## mRNA decay contributes to a reset of the transcriptome at the M-G1 phase transition

It is evident that the expression of G2/M-specific genes is reduced following the completion of cell division, both through scheduled protein degradation and transcriptional inactivation (*Figure 3A*; *Bar-Joseph et al., 2008*; *Castro et al., 2005*; *Chaudhry et al., 2002*; *Cho et al., 2001*; *Cho et al., 1998*; *Grant et al., 2013*; *Harper et al., 2002*; *Nakayama and Nakayama, 2006*; *Peters, 2002*; *Vodermaier, 2004*; *Whitfield et al., 2002*). Here, we identify widespread scheduled mRNA decay during mitotic exit and early G1 phase as an additional mechanism acting to reset gene expression following cell division. Why would mRNA transcripts be actively degraded when the clearance of transcripts will eventually be achieved by transcription shut-down alone? All the mRNAs we tested are quite stable during mitosis and late G1 phase (*Figure 3D–E*). Therefore, transcription inhibition by itself would lead to significant carry-over of transcripts into the next cell cycle. Considering the half-live of these mRNAs in G2 and M phase (*Figure 3C*), they would persist in G1 phase for many hours. Therefore, decay-mediated clearance of mRNAs as cells exit mitosis could aid to limit expression of the encoded proteins in G1 phase, especially since most mRNAs that are degraded during the M-G1 phase transition are not translationally repressed in G1 phase (*Tanenbaum et al., 2015*). As these genes include many genes that encode for proteins with important functions in cell cycle control (*Figure 1—figure supplement 1K*), we speculate that their continued expression in G1 phase may perturb normal cell cycle progression, and potentially could even contribute to cellular transformation (*García-Higuera et al., 2008*; *Park et al., 2008*; *Sigl et al., 2009*). Thus, scheduled mRNA decay during the cell cycle may be important to restrict gene expression of many cell cycle genes to their appropriate cell cycle phases.

## CNOT1 promotes two waves of mRNA decay during the M-G1 phase transition

We have identified two consecutive waves of mRNA decay as cells progress through mitosis and into G1 phase (*Figure 2A and C* and *Figure 3—figure supplement 1H*). The fact that mRNA decay occurs in two waves may indicate the existence of two distinct mechanisms that act consecutively to degrade transcripts. Interestingly, these two waves of mRNA degradation during the M-G1 phase transition are highly reminiscent of the two consecutive waves of protein degradation that occur at the M-G1 transition (*Alfieri et al., 2017*; *Sivakumar and Gorbsky, 2015*).

The regulation of mRNA decay often occurs through (sequence) specific interactions between mRNAs and RNA binding proteins (RBPs). Through direct interactions with mRNAs, RBPs can recruit components of the RNA decay machinery, such as the CCR4-NOT complex, to the mRNA. Recruitment of CCR4-NOT, a key regulator of gene expression, will then induce deadenylation of the target transcript, generally followed by degradation (*Garneau et al., 2007*). Interestingly, a previous report

found that the poly(A) tail lengths of the genes we identified as *immediate decay* are already short-ened in early mitosis (*Figure 4B*; *Park et al., 2016*). The observation that poly(A) tails of transcripts decayed during the M-G1 phase transition are shorter in mitosis could suggest that CCR4-NOT-dependent deadenylation during early mitosis marks these transcripts for decay during mitosis and early G1 phase. Indeed, we identified CNOT1, an essential member of the CCR4-NOT complex, as a regulator of post-mitotic mRNA decay (*Figure 4A and D*). We note that the effects of CCR4-NOT depletion on mRNA decay are modest in our experiments, but the magnitude of the effect is likely caused, at least in part, by the inability to effectively deplete CNOT1, while maintaining cells in a cycling state. Nonetheless, it is possible that additional mechanisms contribute to mRNA decay during the M-G1 phase transition. Such mechanisms could involve PARN-dependent deadenylation, or may be independent of mitotic deadenylation and instead rely on mRNA decapping or endonucleolytic cleavage of the mRNA. Rapid inducible degradation systems (*Banaszynski et al., 2006*; *Nishimura et al., 2009*; *Yesbolatova et al., 2020*) could be applied to quickly and potently inhibit CNOT1 and other mRNA decay factors such as DCP2 and XRN1 to shed more light on this question in the future.

Our data shows that both waves of mRNA decay during the M-G1 phase transition are stimulated by CNOT1 (*Figure 4D*). Nonetheless, these waves of mRNA decay may be regulated independently, involving distinct RBPs. Binding of distinct RBPs could ensure the timely decay of specific sets of mRNA during either mitotic exit or early G1 phase. It will be interesting to investigate which RBPs are involved in recognizing different subsets of mRNAs that need to be degraded during particular times in the cell cycle. Identification of such RBPs will allow a better understanding of the function and mechanisms of scheduled mRNA degradation during the cell cycle.

# Materials and methods

## Key resources table

| Reagent type (species) or resource | Designation | Source or reference | Identifiers | Additional information |
|---|---|---|---|---|
| Cell line (*Homo sapiens*) | hTERT RPE-1 | ATCC | CRL-4000 | Cell line maintained in DMEM/F12 |
| Cell line (*Homo sapiens*) | HEK293T | ATCC | CRL-3216 | Cell line maintained in DMEM |
| Antibody | Anti-Histone three phospho-serine 10 (pH3 Ser10) (Rabbit polyclonal) | Upstate | 06–570 | FACS (1:500) |
| Recombinant DNA reagent | pMD2.G | Addgene #12,259 | | Lentiviral packaging plasmid |
| Recombinant DNA reagent | psPAX2 | Addgene #12,260 | | Lentiviral packaging plasmid |
| Recombinant DNA reagent | mkO2-hCdt1(30/120) | *Sakaue-Sawano et al., 2008* | | FUCCI-G1 marker |
| Recombinant DNA reagent | mAG-hGem(1/110) | *Sakaue-Sawano et al., 2008* | | FUCCI-G2 marker |
| Recombinant DNA reagent | dCas9-BFP-KRAB | *Jost et al., 2017* | | CRISPRi construct |
| Recombinant DNA reagent | CRISPRia-v2 with sgRNA non-targeting control | Addgene plasmid #84832 and this paper | sgRNA | GCTGCGCTCCGAGCAACCAC |
| Recombinant DNA reagent | pCRISPRia-v2 with sgRNA CNOT1 #1 | Addgene plasmid #84832 and this paper | sgRNA | GCTCCGGGAAACGCTTCCAG |
| Recombinant DNA reagent | CRISPRia-v2 with sgRNA CNOT1 #2 | Addgene plasmid #84832 and this paper | sgRNA | GCGGAGCTCTAGGGAGTGAG |
| Recombinant DNA reagent | CRISPRia-v2 with sgRNA CNOT1 #3 | Addgene plasmid #84832 and this paper | sgRNA | GCGGAGCTCTAGGGAGTGAG |
| Sequence-based reagent | siRNA luciferase | This paper | sgRNA | CGUACGCGGAAUACUUCGAUU |

*Continued on next page*

*Continued*

| Reagent type (species) or resource | Designation | Source or reference | Identifiers | Additional information |
|---|---|---|---|---|
| Sequence-based reagent | CDK1 qPCR For | This paper | qPCR primers | CTATCCCTCCTGGTCAGTACATGG |
| Sequence-based reagent | CDK1 qPCR Rev | This paper | qPCR primers | CTCTGGCAAGGCCAAAATCAGCCAG |
| Sequence-based reagent | TOP2A qPCR For | This paper | qPCR primers | GTCTCTCAAAAGCCTGATCCTGCC |
| Sequence-based reagent | TOP2A qPCR Rev | This paper | qPCR primers | GTCATCACTCTCCCCCTTGGATTTC |
| Sequence-based reagent | UBE2C qPCR For | This paper | qPCR primers | GATGTCAGGACCATTCTGCTCTCC |
| Sequence-based reagent | UBE2C qPCR Rev | This paper | qPCR primers | GCTCCTGGCTGGTGACCTGC |
| Sequence-based reagent | FBXO5 qPCR For | This paper | qPCR primers | GATCCTAGAAGATGATAAGGGGG |
| Sequence-based reagent | FBXO5 qPCR Rev | This paper | qPCR primers | CACCTTGATTGGATAACTTGGTT |
| Sequence-based reagent | FZR1 qPCR For | This paper | qPCR primers | GCACGCCAACGAGCTGGTGAGC |
| Sequence-based reagent | FZR1 qPCR Rev | This paper | qPCR primers | CAGACACAGACTCCCACTTTACC |
| Sequence-based reagent | ARL6IP1 qPCR For | This paper | qPCR primers | CTACCTTGTTCCCATTCTAGCGCC |
| Sequence-based reagent | ARL6IP1 qPCR Rev | This paper | qPCR primers | GGCGTTTCCACCAACCCACAGC |
| Sequence-based reagent | CENPA qPCR For | This paper | qPCR primers | GCCCTATTGGCCCTACAAGAGGC |
| Sequence-based reagent | CENPA qPCR Rev | This paper | qPCR primers | GGCTCTGGAGAGTCCCCGG |
| Sequence-based reagent | PSD3 qPCR For | This paper | qPCR primers | CTTAAAACTGCCGACTGGAGGGTC |
| Sequence-based reagent | PSD3 qPCR Rev | This paper | qPCR primers | CTTCAGTTGCTCCTCCTGAGACAG |
| Sequence-based reagent | UBALD2 qPCR For | This paper | qPCR primers | CGGCCGACCAGGCGAAGCAG |
| Sequence-based reagent | UBALD2 qPCR Rev | This paper | qPCR primers | CAGCGCATCGGGGAAGTTGGG |
| Sequence-based reagent | SRGAP1 qPCR For | This paper | qPCR primers | GGCAGCCTGACCAACATCAGCCG |
| Sequence-based reagent | SRGAP1 qPCR Rev | This paper | qPCR primers | GGGGCATGCTTTGCTGTGCTCTG |
| Sequence-based reagent | CNOT1 qPCR For | This paper | qPCR primers | GTAGTGCCCTTTGTTGCCAAAG |
| Sequence-based reagent | CNOT1 qPCR Rev | This paper | qPCR primers | GGAGGTTTCCAGGTTTTAGCTC |
| Sequence-based reagent | CDKN1A qPCR For | This paper | qPCR primers | GCACCTCACCTGCTCTGCTGC |
| Sequence-based reagent | CDKN1A qPCR Rev | This paper | qPCR primers | CCTCTTGGAGAAGATCAGCCGG |

*Continued on next page*

*Continued*

| Reagent type (species) or resource | Designation | Source or reference | Identifiers | Additional information |
|---|---|---|---|---|
| Sequence-based reagent | Alt-R crRNA targeting p53 | This paper; ordered from Integrated DNA Technologies (IDT) | crRNA | UCGACGCUAGGAUCUGACUG |
| Sequence-based reagent | Alt-R tracrRNA | Integrated DNA Technologies (IDT) | tracrRNA | |
| Sequence-based reagent | sgRNA non-targeting control | This paper | sgRNA | GCTGCGCTCCGAGCAACCAC |
| Sequence-based reagent | sgRNA CNOT1 #1 | This paper | sgRNA | GCTCCGGGAAACGCTTCCAG |
| Sequence-based reagent | sgRNA CNOT1 #2 | This paper | sgRNA | GCGGAGCTCTAGGGAGTGAG |
| Sequence-based reagent | sgRNA CNOT1 #3 | This paper | sgRNA | GCGGAGCTCTAGGGAGTGAG |
| Sequence-based reagent | siRNA luciferase | This paper | sgRNA | CGUACGCGGAAUACUUCGAUU |
| Sequence-based reagent | siRNA CNOT1 | Dharmacon ON-TARGET plus | siRNA pool of 4 | |
| Sequence-based reagent | smFISH probe for TOP2A | ThermoFisher | VA1-14609 | Fluorophore: Alexa Fluor 546 Sequence: Proprietary |
| Sequence-based reagent | smFISH probe for CDK1 | ThermoFisher | VA6-17545 | Fluorophore: Alexa Fluor 647 Sequence: Proprietary |
| Chemical compound, drug | Propidium Iodide | Sigma-Aldrich | P4170 | |
| Chemical compound, drug | Taxol (Paclitaxel) | Sigma-Aldrich | T1912 | |
| Chemical compound, drug | RO-3306 | Calibochem | 217699 | |
| Chemical compound, drug | Actinoymcin D | Sigma-Aldrich | A9415 | |
| Chemical compound, drug | Hoechst 33,342 | ThermoFisher | H3570 | |
| Chemical compound, drug | TriSure | Bioline | BIO-380032 | |
| Commercial assay or kit | Bioscript Reverse Transcriptase Kit | Bioline | BIO-27036 | |
| Commercial assay or kit | SYBR-Green Supermix | Bio-Rad | #1708880 | |
| Commercial assay or kit | viewRNA smFISH kit | ThermoFisher | QVC0001 | |
| Software, algorithm | Matlab | Mathworks | | |
| Software, algorithm | R | R | | |
| Other | DAPI stain | ThermoFisher | D1306 | 1 µg/ml |
| Other | Wheat Germ Agglutinin | ThermoFisher | W11261 | 1 µg/ml |

## Cell culture

HEK293T cells were maintained in Dulbecco's modified Eagle medium (DMEM) supplemented with 5% fetal bovine serum (FBS, Sigma-Aldrich) and 1% penicillin/streptomycin (Gibco). RPE-1 cells and

derivatives were maintained in DMEM/Nutrient Mixture F-12 (DMEM/F12, Gibco) supplemented with 10% FBS and 1% penicillin/streptomycin. RPE-1 cells were obtained from ATCC, and are not part of the commonly misidentified cell lines. The RPE-1 cells used are all free of mycoplasm.

## Transfections

Plasmid transfections were performed using FuGENE HD (Promega) according to the manufacturer's protocol. Cas9 RNPs were transfected using RNAiMAX (Invitrogen). In short Cas9 loaded with duplexed tracrRNA and crRNAs targeting TP53 (all from Integrated DNA Technologies [IDT]) were transfected according to the manufacturer's protocol. siRNAs were transfected at a final concentration of 10 nM using RNAiMAX according to the manufacturer's protocol. For microscopy, cells were grown and transfected in 96-well microscopy plates (Matriplate, Brooks). For RT-qPCR, cells were grown and transfected in 96-well culture in late (Greiner Bio-One). For FACS analysis of the cell cycle, cells were grown and transfected in six-well culturing plate (Greiner Bio-One). Two days post-transfection, cells were either fixed for smFISH, the RNA was harvested for RT-qPCR analysis, or the cells were dissociated and resuspended in ice-cold PBS for FACS analysis. For knockdown of CNOT1 we used ON-TARGET plus siRNAs from Dharmacon. As a control, we used a custom siRNA targeting luciferase (5'- CGUACGCGGAAUACUUCGAUU-3') from Dharmacon.

## Lentivirus production

Lentivirus was produced by transfecting HEK293T cells with packaging plasmids (pMD2.G and psPAX2; Addgene #12,259 and #12260, respectively) and lentiviral plasmids carrying the transgene of interest. Two days post-transfection, virus was harvested by collecting the culture medium, pelleting cell debris by centrifugation, and collecting the supernatant.

## Generation of cell lines

To generate RPE-FUCCI cells, RPE-1 cells were transduced with lentivirus expressing mkO2-hCdt1(30/120) (FUCCI-G1) and lentivirus expressing mAG-hGem(1/110) (FUCCI-G2) (*Sakaue-Sawano et al., 2008*). Single clones were isolated by FACS on a BD FACSFUSION system. One clone was selected that showed cyclic expression of both reporter constructs. To generate RPE-FUCCI CRISPRi cells, RPE-FUCCI cells were transduced with lentivirus carrying dCas9-BFP-KRAB (*Jost et al., 2017*), and the 15% highest BFP-positive cells were isolated by FACS. RPE-FUCCI Δp53 cells were generated by transfecting Cas9 protein loaded with tracrRNA and crRNAs targeting p53 into RPE-FUCCI cells. Knock-out cells were selected by treatment with Nutlin-3a (Cayman chemical) for 7 days.

## Irradiation

For irradiation, cells were placed into the irradiation chamber of a Gammacell Exactor (Best Theratronics) equipped with a $^{137}$Cs source, and irradiated with the indicated doses of γ-irradiation.

## Synchronization of cells in mitosis

In order to synchronize cells in mitosis, we first arrested cells in G2 by treating cells with the CDK1-inhibtor RO-3306 (10 μM, Calbiochem) for 16 hr. Subsequently, RO-3306 was removed and the cells were washed twice with PBS before applying fresh medium supplemented with Taxol, which blocks cells in mitosis. Finally, 45 min after Taxol addition, mitotic cells were separated from the interphase cells by shaking of the culture dish (shake-off). This specifically detaches mitotic cells, which were then harvested by collecting the culture medium. For some experiments mitotic cells were isolated through shake-off from asynchronously growing populations without pre-treatment with RO-3306 and Taxol, which is indicated in the corresponding figure legends.

## Transcription inhibition

To inhibit transcription, we treated cells with 1 μg/ml Actinomycin D (Sigma-Aldrich) for the indicated durations.

## FACS analysis

To visualize the cell cycle distribution of RPE-FUCCI cells, cells were dissociated, resuspended in ice-cold PBS and analyzed by FACS. To visualize and quantify the DNA content of RPE-FUCCI cells, cells

were incubated with 2 µM of Hoechst 33,342 (ThermoFisher) for 30 min. Cells were then dissociated, resuspended in ice-cold PBS and analyzed by FACS. Cell cycle phases were gated based on FUCCI fluorescence (*Figure 1C*). FlowJo software was used to quantify the mean Hoechsts 33,342 fluorescence intensity of cells within each FUCCI gate. In order to identify mitotic cells, cells were fixed in 80% ethanol (–20°C). Thereafter, cells were stained using an antibody targeting the mitosis-specific marker phosphorylated histone 3 (4N pH3)-ser10 (Upstate, 06–570) and propidium iodide to label DNA content. The mitotic fraction was determined as the fraction of 4N pH3-ser10-positive cells.

## CRISPRi

For CRISPRi, RPE-FUCCI CRISPRi cells were infected with lentivirus particles expressing a non-targeting sgRNA, or an sgRNA targeting CNOT1 (*Horlbeck et al., 2016*) and a puromycin resistance cassette followed by BFP. Two days post-infection, infected cells were selected with puromycin (10 µg/ml) for 3 days to eliminate uninfected cells. Sequences of sgRNAs used in this study can be found in *Supplementary file 3*.

## FACS-isolation of cells in different stages of the cell cycle for qPCR

RPE-FUCCI cells were collected by trypsinization and subsequently resuspended as single cells in PBS supplemented with 0.5% FBS. Cells were sorted using a BD FACSFUSION system. To isolate cells at different moments during G1, we measured the average FUCCI-G1 expression in early S phase cells (for gating strategy, see *Figure 2—figure supplement 1A* and *Supplementary file 1*). Using the FUCCI-G1 fluorescence relative to early S phase, we calculated the upper and lower bounds for the FACS-gating strategy to isolate cells at different times throughout G1 phase. To isolate cells in G2/M, we isolated FUCCI-G2-positive cells from asynchronous populations (*Figure 2—figure supplement 1A*). To specifically isolate G2 cells, mitotic cells were first removed from the population by shake-off (*Figure 2—figure supplement 1D*), and FUCCI-G2 cells were isolated (*Figure 2—figure supplement 1E*). To isolate early versus late mitotic cells, mitotic cells were first isolated by shake-off (*Figure 2—figure supplement 1D*). Subsequently, early and late mitotic cells were separated by sorting mitotic cells expressing high versus low levels of the FUCCI-G2 reporter, respectively (*Figure 2—figure supplement 1F*). For each cell cycle fraction, at least 2500 cells were isolated for downstream RT-qPCR analysis. Following FACS isolation the cells were pelleted and resuspended in TriSure lysis buffer, and subsequently stored at –20°C or processed for RNA isolation (see below).

## RT-qPCR

For RT-qPCR analysis, cells were lysed in TriSure (Bioline) and RNA was extracted according to the manufacturer's protocol. First strand synthesis was performed using Bioscript (Bioline). mRNA expression levels were quantified using SYBR-Green Supermix (Bio-Rad) on a Bio-Rad Real-time PCR machine (CFX Connect Real-Time PCR Detection System). Relative mRNA expression levels were calculated using the ΔΔCt method. GAPDH and RPN1 were selected as reference genes for normalization, based on their reported high mRNA stability (*Schwanhäusser et al., 2011*). Importantly, the use of reference genes for the calculation of gene expression using the ΔΔCt method will correct for changes in mRNA abundance after cell division (i.e. the number of transcripts will decrease by 2-fold upon cell division), as the reference genes are also subject to this effect. RT-qPCR primers were designed using Primer3, for sequences see *Supplementary file 3*.

## smFISH

smFISH was performed using viewRNA probes targeting TOP2A (probe# VA1-14609) and CDK1 (probe# VA6-17545) (ThermoFisher). Staining was done according to the manufacturer's protocol. In brief, cells were grown in 96-well microscopy plates (Matriplate, Brooks) and fixed for 30 min using 4% formaldehyde. Then, cells were permeabilized with detergent solution for 5 min at room temperature (RT), and subsequently treated with protease solution for 10 min at RT. To label the RNAs, cells were incubated with probes targeting TOP2A and CDK1 for 3 hr at 40°C. Subsequent probes (preAmplifier, Amplifier, and Label Probe) were incubated for 1 hr at 40°C. Between probe incubations, cells were washed with wash buffer for 3 × 1 min. After the final incubation (with Label Probe), cells were washed and incubated with DAPI (ThermoFisher, D1306) and WGA, conjugated to Alexa Fluor 488 (ThermoFisher, W11261) to label DNA and membranes, respectively.

## Microscopy

For live-cell microscopy, RPE-FUCCI cells were grown on microscopy plates and imaged using a Nikon Ti-E with PFS, equipped with an Andor Zyla 4.2Mpx sCMOS camera, CFI S Plan Fluor ELWD 20× air objective (0.45 NA), a Lumencor SpectraX light source and Chroma-ET filter sets (89401 and 24002). Temperature and $CO_2$ control were provided by an OKO-lab Boldline microscope cage and $CO_2$ controller. Where indicated, DNA was visualized by incubation of cells with SPY650-DNA (Spirochrome) for 2 hr prior to imaging. SPY650-DNA was resuspended in DMSO according to manufacurer's protocol and used at a 4000-fold dilution. Image analysis was performed using ImageJ software.

For imaging of smFISH stained samples, we used a Nikon TI2 inverted microscope with a perfect focus system, equipped with a Yokagawa CSU-X1 spinning disc, a 100× oil objective (1.49 NA),a Prime 95B sCMOS camera (Photometrics), and a Chroma filter set (ZET405/488/565/640).

## Crystal violet staining

1500 RPE-FUCCI cells (wild-type or p53 knock-out) were seeded per well in six-well dishes. Four hours after seeding 5 µM Nutlin-3a (Cayman chemical) or DMSO were added to indicated wells. After 7 days, cells were washed with PBS and fixed with 100% MeOH for 10 min. Cells were then washed with PBS and incubated with 1.5% crystal violet overnight. The next morning, crystal violet was removed and the plates were washed three times with water and air-dried.

## Single Cell RNA-Seq of FACS-Sorted Cells (SORT-seq)

scRNA-seq of FACS-sorted cells (SORT-seq) was performed as described previously (*Muraro et al., 2016*). Briefly, we sorted in total 104 G2/M phase cells (FUCCI-G1-negative and FUCCI-G2-positive cells, *Figure 1C*) and 893 G1 phase cells (FUCCI-G1-positive and FUCCI-G2-negative cells, *Figure 1C*) as single cells in three 384-well plates. After sorting, these cells were subjected to scRNA-seq based on the CEL-Seq2 protocol (*Hashimshony et al., 2016*). scRNA-seq was performed by Single Cell Discoveries (https://www.scdiscoveries.com). After sequencing, we continued with cells (841 in total) that passed quality tests (we removed cells with less than 5900 UMIs or more than 111.000 UMIs to lose low-quality cells and doublets, respectively). Finally, we normalized for differences in mRNA recovery per cell using Monocle2 (R package). Normalization of mRNA recovery corrects for the 2-fold decrease of mRNA content that occurs as a consequence of cell division. Each 384-well plate contained G1 phase cells from 0 to 4 hr after the start of G1 phase, but only one plate contained G1 phase cells from 4 to 9 hr after the start of G1 phase. Therefore, we only used cells from 0 to 4 hr after the start of G1 phase to identify differentially expressed genes. In subsequent analyses (i.e. the spline analysis and the modeling) we did use all G1 phase cells.

## Cell cycle timing using the FUCCI system

To obtain a temporal transcriptome profile as cells progress from mitosis into G1 phase, we wanted to compute a cell cycle time for each sorted G1 phase cell (i.e. how much time a cell has spent in G1 phase at the moment of sorting). Since FUCCI-G1 levels positively correlate with the amount of time a cell has spent in G1 phase, we reasoned that we could use the measured FUCCI-G1 levels to infer a cell cycle time for a G1 phase cell. To characterize precisely how FUCCI-G1 levels increase during G1 phase, we imaged RPE-FUCCI cells under the microscope with a time interval of 5 min and selected cells that progressed through mitosis into G1 phase. Next, we measured the mean fluorescence intensities of both FUCCI sensors in a region of interest (ROI) in the nucleus using ImageJ and subtracted background signal measured in an extracellular ROI. In each experiment we quantified the fluorescence intensities of the FUCCI sensors in 30 cells.

To compute the average FUCCI-G1 time trace during G1 phase, we aligned the time traces of individual cells at the metaphase-to-anaphase transition, which is defined by a sudden decrease in FUCCI-G2 fluorescence (*Sakaue-Sawano et al., 2008*). Next, since the total amount of time a cell spends in G1 phase differs for each cell, we clipped individual time traces at the end of G1 phase, which ends shortly after the FUCCI-G2 levels start to increase (*Grant et al., 2018*). To determine the moment the FUCCI-G2 levels start to increase, we first corrected the FUCCI-G2 traces for fluorescence crosstalk from the FUCCI-G1 marker, which is also excited by the 488 nm laser used for imaging of the FUCCI-G2 marker. To correct the FUCCI-G2 time traces for crosstalk from the FUCCI-G1 marker, we subtracted at each time point 31% of the FUCCI-G1 fluorescence intensity from the FUCCI-G2

fluorescence intensity. Next, we determined the time point when the mean FUCCI-G2 fluorescence intensity reached a threshold value, which was set by visual inspection, and clipped all FUCCI-G1 time traces at the time point of FUCCI-G2 increase. Finally, we computed for each experiment the average FUCCI-G1 levels from the moment of metaphase and fit the average of three experiments to a third-order polynomial (*Figure 1D*).

To directly compare the FUCCI-G1 levels that we measured on the microscope to the FUCCI-G1 levels that we measured on the FACS, we normalized both microscopy- and FACS-measured FUCCI-G1 levels to the average FUCCI-G1 level of early S phase cells. To quantify the mean FUCCI-G1 fluorescence intensity in early S phase cells on the microscope, we analyzed the mean nuclear intensities of at least 700 cells per experiment (*Figure 1—figure supplement 1D*). As above, we compensated for fluorescence crosstalk from the FUCCI-G1 marker into the FUCCI-G2 channel by subtracting 31% of the FUCCI-G1 fluorescence intensity from the FUCCI-G2 fluorescence intensity. Next, we determined the range of fluorescence intensities for both the FUCCI-G1 and FUCCI-G2 markers (by subtracting the lowest fluorescence intensity from the highest fluorescence intensity), and defined the early S phase population as those cells with FUCCI-G2 intensities between 2.5% and 10% of the range of FUCCI-G2 intensities and FUCCI-G1 intensities higher than 2.5% of the range of FUCCI-G1 intensities (*Figure 1—figure supplement 1D*, yellow dots). We computed the average FUCCI-G1 level of the early S phase cells, and normalized the third-order polynomial fit against the average FUCCI-G1 level of the early S phase cells.

To quantify the mean FUCCI-G1 fluorescence intensity in early S phase cells on FACS, we analyzed the FUCCI sensors on FACS (*Figure 1C*). We define the early S phase population as those cells that have high FUCCI-G1 levels and have started to increase the expression of the FUCCI-G2 marker (*Figure 1C*), and computed the average FUCCI-G1 level in early S phase cells. To obtain a cell cycle time for each G1 phase cell that was sequenced, we determined the FUCCI-G1 fluorescence intensity level that was obtained by FACS and normalized the FUCCI-G1 level to the average early S phase FUCCI-G1 value. Finally, we used the third-order polynomial fit to infer the cell cycle time of each G1 phase cell from its normalized FUCCI-G1 level.

Because of cell-to-cell variability in FUCCI-G1 fluorescence during G1 phase (*Figure 1—figure supplement 1B*), the cell cycle times computed for individual G1 phase cells based on their FUCCI-G1 fluorescence intensity is an approximate cell cycle time, which is based on the average FUCCI-G1 fluorescence intensity of many cells. To estimate the error in the cell cycle time caused by this cell-to-cell heterogeneity in FUCCI fluorescence intensities, we first determined at every time point the mean and standard deviation (SD) of all 90 FUCCI-G1 fluorescence intensity traces. Next, for each time point we calculated the mean intensity +1 or –1 SD. Using the intensity values of +1 and –1 SD, we calculated the cell cycle time using our polynomial model (*Figure 1D*). Next, we computed the difference in cell cycle time between the calculated cell cycle times for +1 and –1 SD intensity values and the cell cycle time for the mean intensity value. Finally, we determined the average time difference of the +1 and –1 SD and refer to this time as the SD of FUCCI-G1 timing.

## Cell cycle timing using Monocle2

To rank cells using Monocle2 (R package), we used all G2/M phase cells and G1 phase cells that were from the first 4 hr of G1 phase (based on FUCCI cell cycle timing; see section *SORT-seq*). Next, we performed an initial differential transcriptome analysis comparing G2/M phase (FUCCI-G1 marker negative and FUCCI-G2 marker positive) and G1 phase cells (FUCCI-G1 marker positive and FUCCI-G2 marker negative) to select differentially expressed genes that Monocle2 can use in subsequent steps to reconstruct the single-cell trajectory. Monocle2 selected a total of 430 genes that were used to reconstruct the single-cell trajectory, and both the cell rank- and the Monocle2-assigned pseudo times were compared to FUCCI-based ranking and cell cycle timing.

## Differential transcriptome analysis

Differential transcriptome analysis was performed with Monocle2 (R package), either using FUCCI-based or Monocle2-based cell cycle time. For the differential transcriptome analysis (both for the FUCCI-based and the Monocle2-based analysis) we used all G2/M phase cells and only G1 phase cells from the first 4 hr of G1 phase (see section *SORT-seq*). To increase the confidence of our differential transcriptome analysis, we only selected genes for analysis that were clearly detected in all three

384-well plates. To select detected genes, we computed for each gene in each single 384-well plate its average expression in G2/M phase cells (as we didn't want to bias against genes that were down-regulated in G1 phase), and only selected genes which had an average expression of at least two reads in each single 384-well plate. This resulted in a dataset of 3985 genes. Finally, after differential transcriptome analysis, genes that showed at least a 2-fold increase or decrease in expression and had at least a p-value of 1.2547E$^{-5}$ (based on a Bonferroni correction from a p-value of 0.05) were selected as upregulated or downregulated genes, respectively.

## Spline analysis

For the spline analysis (performed in Matlab R2018b), we used the full set of 841 cells (see section *SORT-seq*). We selected the 220 genes that were identified in the differential transcriptome analysis as downregulated in G1 phase, and fit each gene profile with a smoothing spline. Next, we computed the derivative of the splines at each time point and determined the time when the derivative was minimal for each gene (i.e. the moment mRNA levels decreased most). To compare different genes to each other, we normalized the derivative of each gene to its minimum value (i.e. setting the minimum value to –1). Finally, we determined for each gene the first time point during which the normalized derivative was at least –0.95 (where –1.0 is the minimum slope after normalization), and divided genes into two groups; one group in which the minimum slope was reached at the first time point (i.e. during mitosis) and one group in which the minimum slope was reached during G1.

## Calculation of transcription rates

To assess transcription rates in different cell cycle phases of the set of genes that is downregulated in G1 phase, we made use of a previously published dataset (*Battich et al., 2020*) (accession number GSE128365, RPE-1 labeled and spliced dataset). In the experiment used to create this dataset, RPE-FUCCI cells were incubated for varying amount of times with EU, an analog of uridine. Transcripts containing EU were biotinylated and separated from transcripts without EU, using streptavidin magnetic beads. Using a sc-seq pipeline (scEU-seq, *Battich et al., 2020*), the amount of labeled and unlabeled transcripts was determined in single cells. For our analysis, we used data of cells that were incubated for 30 min in EU (i.e. those cells whose 'Condition_Id' is listed as 'Pulse_30' in the meta-data). We identified cells in G1 phase (i.e those cells whose 'Cell_cycle_relativePos' is between 0.00 and 0.33, see *Battich et al., 2020*) and cells in G2 phase (i.e. those cells whose 'Cell_cycle_relativePos' is between 0.83 and 1.00, see *Battich et al., 2020*). For each of the 220 downregulated genes, we computed the average number of labeled transcripts in G1 phase cells and in G2 phase cells and we computed the ratio of labeled transcripts in G1 phase compared to G2 phase (see *Figure 3A* and *Figure 3—figure supplement 1A*). We excluded genes for which the average number of labeled transcripts in G2 phase is zero.

Modeling mRNA decrease mRNA levels (m) depend on the synthesis (µ) and degradation (γ) rate, and the change in mRNA levels over time can be described as follows:

$$\frac{dm}{dt} = \mu - \gamma \cdot m \tag{1}$$

To describe the mRNA levels as cells progress from mitosis into G1 phase, we assumed a simple model in which the observed decrease of mRNA levels is explained by a decrease in the synthesis rate and/or an increase in the degradation rate at a specific time point during M or early G1 phase (referred to as the onset time or $t_{onset}$). When mRNA levels start at a given value ($m_0$), the solution of *Equation 1* results in the following expression for the mRNA levels over time.

$$m(t) = \frac{\mu}{\gamma} + (m_0 - \frac{\mu}{\gamma}) \cdot e^{-\gamma t} \tag{2}$$

Furthermore, we assumed that mRNA levels remain constant before the onset time, resulting in the following pair of equations to describe the mRNA levels as cells progress from mitosis into G1 phase.

$$m(t) = m_0 \qquad\qquad t < t_{onset}$$
$$m(t) = \frac{\mu}{\gamma} + (m_0 - \frac{\mu}{\gamma}) \cdot e^{-\gamma(t - t_{onset})} \quad t < t_{onset} \tag{3}$$

For each gene, we optimized $t_{onset}$ (performed in Matlab R2018B) using an iterative search (between 0 and 370 min after metaphase in steps of 10 min), in which we optimized $m_0$, $\mu$, and $\gamma$ using least square fitting for each $t_{onset}$. Finally, we computed a sum of squared errors (SSE) between the data (using the full dataset of 841 cells) and model for each $t_{onset}$ and selected the $t_{onset}$ with the minimal SSE.

### Calculating half-lives

We computed mRNA half-lives from the degradation rates ($\gamma$) (that we obtained from the modeling) using *Equation 4*.

$$\text{Half-life} = \frac{\ln 2}{\gamma} \tag{4}$$

### Statistics

Statistical comparisons were made using an unpaired one-tailed Student's t-test (*Figures 2E, F, 3C, D, E, 4A, B and C*, *Figure 2—figure supplement 1K-L*, *Figure 3—figure supplement 1A and M*, and *Figure 4—figure supplement 1G*), a paired one-tailed Student's t-test (*Figure 3B*) or a one-tailed Welch's t-test (*Figure 4D*).

## Acknowledgements

We thank members of the Tanenbaum group for helpful discussions, and Xiaowei Yan for help during the initial stages of the project. We thank the Hubrecht Institute flow cytometry facility and single-cell sequencing facility (now Single Cell Discoveries) for their technical support. This work was financially supported by the European Research Council (ERC) through an ERC starting grant (ERCSTG 677936-RNAREG) to MET; MET is also supported by the Oncode Institute that is partially funded by the Dutch Cancer Society (KWF).

## Additional information

### Funding

| Funder | Grant reference number | Author |
|---|---|---|
| European Research Council | ERCSTG 677936-RNAREG | Marvin E Tanenbaum |

The funders had no role in study design, data collection and interpretation, or the decision to submit the work for publication.

### Author contributions

Lenno Krenning, Conceptualization, Data curation, Formal analysis, Investigation, Methodology, Validation, Visualization, Writing – original draft; Stijn Sonneveld, Conceptualization, Data curation, Formal analysis, Investigation, Software, Validation, Writing – original draft; Marvin E Tanenbaum, Conceptualization, Data curation, Funding acquisition, Investigation, Methodology, Project administration, Resources, Supervision, Writing – review and editing

### Author ORCIDs

Lenno Krenning http://orcid.org/0000-0001-6696-0512
Marvin E Tanenbaum http://orcid.org/0000-0001-8762-0090

### Decision letter and Author response

Decision letter https://doi.org/10.7554/eLife.71356.sa1
Author response https://doi.org/10.7554/eLife.71356.sa2

## Additional files

### Supplementary files

• Supplementary file 1. Analyses used in this study. This file contains analyses that were used

throughout the study, in *Figure 1D–E*, *Figure 1—figure supplement 1J,L,N*, *Figures 2A–B and 3C* and *Figure 3—figure supplement 1B-G*.

• Supplementary file 2. Sample size indication. This file contains sample sizes for all experiments that involved single-cell analyses.

• Supplementary file 3. Nucleotide sequences. This file contains nucleotide sequences for reagents that were used in this study; RT-qPCR primer sequences and sgRNA sequences.

• Source data 1. Single-cell transcript counts plate 1.

• Source data 2. Single-cell transcript counts plate 2.

• Source data 3. Single-cell transcript counts plate 3.

• Source data 4. Single-cell sequencing metadata.

• Transparent reporting form

### Data availability

Source data containing single cell transcript counts that were used in this study are provided as supplementary data.

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
