## [Editor Report]

This work is a very significant contribution on how mRNA stability regulation takes place across the cell cycle, particularly after mitosis. It also describes an original method for time-resolved transcriptome analysis during the cell cycle, which is potentially very interesting for the whole molecular and cell biology community.

---

## [Decision Letter]

[Editors' note: this paper was reviewed by Review Commons.]

**Decision letter after peer review:**

Thank you for submitting your article "Time-resolved single-cell sequencing identifies multiple waves of mRNA decay during mitotic exit" for consideration by *eLife*. Your article has been reviewed by 3 peer reviewers at Review Commons, and the evaluation at *eLife* has been overseen by a Reviewing Editor and James Manley as the Senior Editor. We have also sought a new reviewer with specific expertise and discussed the reviews with one another. We would like to invite a revised submission, with these comments in mind:

*Reviewer #1:*

The manuscript describes interesting and relevant insights into cell-cycle controlled mRNA decay and in my opinion is suitable for publication in *eLife* after revision. Specifically, I recommend the following adjustments:

1) Include in the Discussion section the possible involvement of other mRNA decay factors in cell-cycle dependent RNA decay, as already indicated in the response to reviewer 1. Given the lack of data demonstrating the biological relevance of a controlled RNA decay after mitosis, the discussion should in my opinion toned down appropriately.

2) The proposed experiments addressing a possible relationship between G0/G1 bifurcation and the described two-wave mRNA decay model will certainly strengthen the manuscript and I highly recommend including such data in a revised version.

3) As acknowledged by the authors in response to reviewer 3, further analyses on concomitant transcription levels of immediate decay and delayed decay genes would certainly help to clarify the overall impact of cell-cycle controlled mRNA decay.

*Reviewer #2:*

I believe the work is potentially suitable for publication in *eLife* but additional data is needed to support the major claims.

This paper essentially makes three major points:

Description of a new methodology of combining FUCCI sensor levels with sc-RNA-seq to get time-resolved RNA dynamics.

• For this main part of the manuscript, I think the authors need to do a few more experiments to fully validate this method, to determine the resolution and level of accuracy it provides and as Reviewer #4 states "to improve reproducibility" by other labs trying to replicate their results.

• In their rebuttal, the authors have already included some additional analysis that helps differentiate G0 and G1 cells. They also propose to perform new experiments in cells lacking p53 which should be incapable of entering G0. These would be nice additions to the manuscript and would address some of my comments.

• Further characterization of the accuracy and time resolution is also needed in order to correctly interpret their results. There is variability in the expression levels of the FUCCI sensors independent of time. Some sort of quantification of this is needed to understand the error in their time estimates.

• These additions should be relatively straight forward and would improve the manuscript.

mRNA is decayed in two waves after mitosis

• The authors have mostly addressed my comments on this point and have proposed several experiments to further address these comments including additional validation of FBXO5 and to look at mRNA levels at a higher time resolution within mitosis (telophase, metaphase, as well as anaphase).

• I also agree with Reviewer #4 that the author's comments about the importance of these waves of mRNA decay are not supported by any experiments. The authors should either edit the text to be more conservative or perform additional experiments to support this claim.

CNOT1 contributes to the observed mRNA decay after mitosis

• The data supporting this conclusion are relatively weak. The change in mRNA decay upon CNOT1 knockdown are all relatively minor. While the changes seem to be statistically significant, it’s not clear if these changes are large enough to be biologically relevant. They also don't appear robustly repeatable as CDK1 mRNA is rescued in Figure 4A but not in 4D.

• I think the authors should edit the text here to reflect that CNOT1 may contribute somewhat to the decay but that it is not required, as requested by Reviewer 4, and to discuss additional factors that may be involved as requested by Reviewer 3.

In summary, I think the authors have either already addressed or have proposed experiments/analysis to address most of the reviewers comments about the methodology as well as the two waves of mRNA decay. However, the authors do not appear to have a clear plan to try and address the mechanism of the mRNA decay and whether or not CNOT1 is a major regulator of this decay. I think the relatively weak phenotypes with CNOT1 depletion are a major weakness of the manuscript but I recognize that the experiments needed to support their claims may not be feasible in a reasonable time.

*Reviewer #3:*

I think that the paper by Tanenbaum and colleagues is suitable for publication in *eLife* as long as they:

1) Perform the new experiments that they described in the rebuttal letter. Particularly important is to perform measurements of mRNA sysnthesis by addressing nascent transcription rates. The authors propose to include an analysis addressing the relative transcription levels of 'immediate decay' and 'delayed decay' genes. I agree with the authors that these results could clarify most of the concerns in this respect.

2) Carefully revise the text to avoid too strong conclusions not directly related to the results, as pointed out by the fourth referee.

3) Modify the Discussion section in order to consider the possibility that other mRNA decay factors may act in parallel to CNOT1 or in concert with it. This is particularly convenient if we considered the information supplied by the authors in response to Reviewer 1, describing the results of experiments addressing the possible involvement of DCP2 in post-mitotic mRNA decay (Figure 1A,B of the rebuttal letter).

In conclusion, I consider that this manuscript, with the new experiments and changes proposed by the authors, is a significant contribution on how mRNA stability regulation takes place accross the cell cycle, particularly after mitosis. This results should be of interest for a broad audience in the molecular and cell biology community, which would appreciate a paper like this in *eLife*.

*Reviewer #4:*

In this work, Krenning and colleagues aim to investigate the mechanisms controlling periodic cell cycle mRNAs expression in mammalian cells. While previous studies extensively characterized the control of periodic mRNA synthesis, as well as cell cycle regulated protein destruction, here the authors investigate the control of mRNA degradation, which they suggest may contribute to sharp transitions between cell cycle phase and unidirectional cycles. To this end, they combine already existing tools – single-cell sequencing and cell cycle monitoring by using the fluorescent reporter FUCCI (Sakaue-Sawano et al. 2008 PMID: 18267078) – to precisely correlate the cell cycle phase of a cell and its transcriptome.

Here lies the novelty of this work. Previous reports (by the authors and others) described a similar approach, coupling FUCCI, single cell sequencing and metabolic labelling, to measure mRNA synthesis and degradation rates in mammalian cells (e.g. Battich et al. 2020, PMID: 32139547). Using this approach however, the RNA labelling times were long (up to 6 hours for pulse-chase experiments), precluding to capture rapid changes in mRNA expression that may occur during cell cycle phase transitions. To overcome this limitation, Krenning and colleagues set to precisely calibrate the FUCCI reporter fluorescence during the cell cycle of HEK293T cells, using a combination of live imaging and FACS analysis. This fluorescence-based calibration is used to obtain a cell cycle progression (pseudo)time for each cell by FACS. Then single cells are sorted to perform single-cell sequencing, thus providing the significant advantage that cells do not be synchronized before gene expression measurements. Using this tool and a combination of cell cycle synchronization, transcription inhibition and mathematical modelling, the authors conclude that two-waves of mRNA degradation occur for a sub-class of cell cycle mRNAs during the M/G1 phase transition. Furthermore, they show that depletion – even if incomplete – of the decay factor CNOT1 promotes partial stabilization of the mRNAs degraded during the M/G1 phase transition.

While I think that the method is original and it has the potential to address important biological questions, I found two major weakness in this study. First, the accuracy of the FUCCI reporter in determining cell cycle progression (pseudo)time in single cells is not definitively demonstrated. A key aspect that remains unclear is how variable is the expression of the FUCCI reporter and how the calibration suffers from cell-to-cell heterogeneity in the reporter expression and cell cycle duration. This could affect the results of experiments where the reporter is used to sort cells based on the cell cycle stage to measure mRNA expression on pools of cells (e.g. Figure 2C, Figure 3-figure supplement 1A-F). Second, the authors conclude that "mRNA degradation occurs at the M/G1 phase transition, and provides an important contribution to the reset of the transcriptome after cell division" and that "mRNA decay requires CNOT1". I find that these claims are very strong and they are not fully supported by the data presented in this manuscript. For instance no experiment is performed to determine the importance of this decay mechanism and its consequences on cell cycle. Furthermore, the CNOT1 downregulation experiment shows that CNOT1 provides a small, even if significant, contribution to decay, not that it is required.

Overall, the impact of this work on the cell cycle field is two-fold. First, it describes technology that, provided that an accurate calibration is done, can be applied to numerous model organisms since the FUCCI system is available, for instance, for mice, flies and zebrafish (see review by Zielke and Edgar, 2015, PMID: 25827130). Second, this study provides novel insights on the regulation of mRNA degradation during the mammalian cell cycle.

Major concerns:

The authors claim that their method can accurately measure the cell cycle stage and the "absolute" time an individual cell spent in the cell cycle with a resolution in the scale of minutes (p. 5). I have few concerns here, that affect also the strength of the biological conclusions. Here my specific points:

a) The time resolution estimate is based on an average cell cycle length determined by fluorescence microscopy (Figure 1A). The cell-to-cell variability in cell cycle length should be computed and used to estimate the error of the time resolution.

b) The authors combined microscopy and FACS to calibrate the fluorescence measured with the two methods. In particular they base their calibration on the identification of cells in early S phase to normalize the two fluorescence datasets. Thus, accurate identification of early S phase cells seems very important to compare the fluorescence detected with the two methods. It is unclear from the text and methods section whether they used also independent approaches to determine the cell cycle phase. For instance in the FUCCI paper (Sakaue-Sawano et al. 2008 PMID: 18267078) Hoechst staining is used to independently determine the cell cycle. Furthermore, by microscopy, how did the authors determine when cells were undergoing early S-phase? Did they use FUCCI alone? To clarify this point, they could add a fluorescence microscopy video where they indicate how they classify cell cycle stages. This will be very important to ensure reproducibility of the method.

c) The FUCCI reporter expression heterogeneity is not clearly discussed in the text and it is unclear whether it may affect the accuracy of the method. The authors should provide a sample microscopy video (same as in (b) to show how heterogeneous (or not) is the expression of the reporter across many synchronized cells.

d) The biological discovery of this paper is that mRNA levels decline in multiple waves during mitotic exit. I find the text at times confusing. For instance at page 7, the paragraph title "mRNA levels decline in multiple waves after cell division", suggests that both degradation waves occur after completion of cell division. But this is not what is shown in Figure 2D. The "immediate" degraded mRNAs are degraded before the end of mitosis, according to the FISH quantifications. I highly recommend to clarify the text.

e) The method presented in this manuscript provides the ability to precisely estimate the duration of the different cell cycle phases of HEK293 cells. One result that I would recommend adding is the duration of the different cell cycle phase of an average HEK293 cell, based on your FUCCI reporter estimates. This will be important to interpret many of your plots where the X axis is presented as "Time after metaphase". I make this point because I think that some of the mRNA levels declines reported in your study may be explained by a dilution effect occurring upon cell division. I think that this is particularly important for the "delayed" mRNA pool. Could the authors please comment this point in the text? How did they consider dividing cells while performing single cell imaging?

f) FISH is used as in independent mean to measure mRNA reduction in single cells at different stages of the cell cycle. For these experiments, two mRNAs belonging to the "immediate" degraded group are measured, TOP2A and CDK1 and both nicely show a decrease before the end of the cell cycle. Could you add the single cell data to the FISH quantifications shown in figures 2E, 2F, and 4A, to report the cell-to-cell variability observed for the expression of these mRNA?

g) As a follow-up question, why didn't you perform FISH also on two mRNAs from the "delayed decrease" group, to show changes in mRNA expression? This seems an important control if the aim is to demonstrated trough impendent methods that two waves of degradation occur during the M/G1 phase transition.

h) The authors conclude that "mRNA degradation occurs at the M/G1 phase transition, and provides an important contribution to the reset of the transcriptome after cell division". However, no experiment is performed to determine the importance of this decay mechanism and its consequences on cell cycle. For instance it could be tested by selectively downregulating the expression of CNOT during the M/G1 phase transition by using an inducible degron system. I understand that this may be a complicated experiment to do, but if a demonstration is not provided, then the text should remain more conservative in describing the results.

i) The authors conclude that "mRNA decay requires CNOT1". The CNOT1 downregulation experiments show that CNOT1 contributes to decay, not that it is required. This should be changed in the text.

---

## [Author Response]

Reviewer #1:The manuscript describes interesting and relevant insights into cell-cycle controlled mRNA decay and in my opinion is suitable for publication in eLife after revision. Specifically, I recommend the following adjustments:1) Include in the Discussion section the possible involvement of other mRNA decay factors in cell-cycle dependent RNA decay, as already indicated in the response to reviewer 1.

In the Discussion section we now mention several other potential mRNA decay mechanisms that may contribute to cell cycle dependent RNA decay. See page 15, lines 5 to 8:

“Nonetheless, it is possible that additional mechanisms contribute to mRNA decay during the M-G1 phase transition. Such mechanisms could involve PARN-dependent deadenylation, or may be independent of mitotic deadenylation and instead rely on mRNA decapping or endonucleolytic cleavage of the mRNA.”

Given the lack of data demonstrating the biological relevance of a controlled RNA decay after mitosis, the discussion should in my opinion toned down appropriately.

We have toned down our conclusions regarding the biological relevance of RNA decay after mitosis. We have done this at the following locations in the text:

Introduction: Page 4, line 36 to page 5, line 2

“Together, our findings demonstrate that, analogous to protein degradation, mRNA degradation occurs at the M-G1 phase transition, and provides an important contribution to the reset of the transcriptome after cell division” changed to “Together, our findings demonstrate that, analogous to protein degradation, scheduled mRNA degradation occurs at the M-G1 transition. Scheduled mRNA degradation *likely* provides an important contribution to the reset of the transcriptome after cell division”.

Discussion: Page 14, lines 10-11

“Therefore, decay-mediated clearance of mRNAs as cells exit mitosis will aid to limit expression of the encoded proteins in G1 phase” changed to “Therefore, decay-mediated clearance of mRNAs as cells exit mitosis could aid to limit expression of the encoded proteins in G1 phase”.

Discussion: Page 14, lines 13-16

“As these genes include many genes that encode for proteins with important functions in cell cycle control, their continued expression in G1 phase may perturb normal cell cycle progression…” changed to “As these genes include many genes that encode for proteins with important functions in cell cycle control, we speculate that their continued expression in G1 phase may perturb normal cell cycle progression…”.

2) The proposed experiments addressing a possible relationship between G0/G1 bifurcation and the described two-wave mRNA decay model will certainly strengthen the manuscript and I highly recommend including such data in a revised version.

We have included experiments performed in p53 knock-out cells (which prevents cells from entering G0 phase). These experiments demonstrate that mRNA decay still occurs in two independent waves, demonstrating that these two waves of mRNA decay do not represent distinct waves occurring in G0 and G1 phase cells (see new Figure 2—figure supplement 1I-M). We discuss these results in the main text on page 9, lines 12 to 23.

We have analyzed the expression of genes belonging to the *immediate* and *delayed* decay groups in individual cells. This analysis shows that *both* sets of genes are downregulated in the same set of cells (new Figure 2—figure supplement 1N), indicating that both waves of mRNA decay occur in individual cells, irrespective of G1/G0 status. This result is discussed in the main text on page 9, lines 24 to 36.

3) As acknowledged by the authors in response to reviewer 3, further analyses on concomitant transcription levels of immediate decay and delayed decay genes would certainly help to clarify the overall impact of cell-cycle controlled mRNA decay.

We have now included analysis of mRNA synthesis rates for *immediate* and *delayed decay* genes (see Figure 3A and Figure 3—figure supplement 1A). These results indicate that G1 phase transcription shutdown occurs for the large majority of genes that undergo scheduled mRNA decay, demonstrating a dual mechanism of gene silencing for these genes. These data are discussed in the main text on page 10, lines 2 to 9.

Reviewer #2:I believe the work is potentially suitable for publication in eLife but additional data is needed to support the major claims.This paper essentially makes three major points:Description of a new methodology of combining FUCCI sensor levels with sc-RNA-seq to get time-resolved RNA dynamics.• For this main part of the manuscript, I think the authors need to do a few more experiments to fully validate this method, to determine the resolution and level of accuracy it provides and as Reviewer #4 states "to improve reproducibility" by other labs trying to replicate their results.

To address this point, we have now included multiple experiments that test the accuracy of our method.

First, we have quantified the cell-to-cell heterogeneity in FUCCI-G1 fluorescence and the corresponding heterogeneity (i.e. error) in the calculated G1 timing of single cells (see new Figure 1—figure supplement 1C and Methods). This analysis precisely quantifies the accuracy of our timing method at the single cell level. It is important to point out that mRNA decay rates which we report (e.g. figures 3B and 3C, Supplementary file 1) are calculated based on many cells, thus averaging out much of the cell-to-cell heterogeneity in FUCCI fluorescence. Therefore, the calculated mRNA decay rates are expected to be very accurate. We have inserted this new analysis in the manuscript (Figure 1—figure supplement 1C) and we discuss it in the main text (page 5, lines 28 to 31) and in the discussion (page 13, lines 21 to 24).

Second, we have included Hoechst-based analysis of DNA content (new Figure 1—figure supplement 1A), which independently confirms our gating strategy for cells in different phases of the cell cycle. This provides an extra quality check that helps to improve the reproducibility of our FACS-sorting based isolation of cells in different positions along the cell cycle.

Finally, we have included example images of RPE-FUCCI cells as they progress through mitosis and G1/G0 phase (new Figure 1B). We included these images in the section of the text describing the FUCCI system (page 5, lines 15-30). These images provide clear examples of how cells in the different cell cycle phases are identified by microscopy, and as such will also improve the reproducibility of this work by others.

• In their rebuttal, the authors have already included some additional analysis that helps differentiate G0 and G1 cells. They also propose to perform new experiments in cells lacking p53 which should be incapable of entering G0. These would be nice additions to the manuscript and would address some of my comments.

We have included experiments performed in p53 knock-out cells (which prevents cells from entering G0 phase). These experiments demonstrate that mRNA decay still occurs in two independent waves, demonstrating that these two waves of mRNA decay do not represent distinct waves occurring in G0 and G1 phase cells (see new Figure 2—figure supplement 1I-M). We discuss these results in the main text on page 9, lines 12 to 23.

We have analyzed the expression of genes belonging to the *immediate* and *delayed* decay groups in individual cells. This analysis shows that *both* sets of genes are downregulated in the same set of cells (new Figure 2—figure supplement 1N), indicating that both waves of mRNA decay occur in individual cells, irrespective of G1/G0 status. This result is discussed in the main text on page 9, lines 24 to 36.

• Further characterization of the accuracy and time resolution is also needed in order to correctly interpret their results. There is variability in the expression levels of the FUCCI sensors independent of time. Some sort of quantification of this is needed to understand the error in their time estimates.

We agree that this in an important point. As mentioned above, we have quantified the cell-to-cell heterogeneity in FUCCI-G1 fluorescence and the corresponding heterogeneity (i.e. error) in the calculated G1 timing of single cells (see new Figure 1—figure supplement 1C and Methods). This analysis precisely quantifies the accuracy of our timing method at the single cell level. It is important to point out that mRNA decay rates which we report (e.g. figures 3B and 3C, Supplementary file 1) are calculated based on many cells, thus averaging out much of the cell-to-cell heterogeneity. Therefore, the calculated mRNA decay rates are expected to be very accurate. We have inserted this new analysis in the manuscript (Figure 1—figure supplement 1C) and we discuss it in the main text (page 5, lines 28 to 31) and in the discussion (page 13, lines 21 to 24).

• These additions should be relatively straight forward and would improve the manuscript.mRNA is decayed in two waves after mitosis• The authors have mostly addressed my comments on this point and have proposed several experiments to further address these comments including additional validation of FBXO5 and to look at mRNA levels at a higher time resolution within mitosis (telophase, metaphase, as well as anaphase).

We have now included additional analysis of the expression levels of 5 ‘immediate’ and 5 ‘delayed’ decay genes in G2 phase compared to early / late mitosis using qPCR. These data are included in the manuscript (Figure 2—figure supplement 1G-H), and demonstrate that the levels of ‘immediate decay’ genes decrease as cells move from early to late mitosis, while ‘delayed’ decay gene expression does not decrease until G1 phase. We discuss these results in the main text at lines page 8, line 33 to page 9, line 11.

• I also agree with Reviewer #4 that the author's comments about the importance of these waves of mRNA decay are not supported by any experiments. The authors should either edit the text to be more conservative or perform additional experiments to support this claim.

We have toned down our conclusions regarding the biological relevance of RNA decay after mitosis. We have done this at the following locations in the text:

Introduction: Page 4, line 36 to page 5, line 2

“Together, our findings demonstrate that, analogous to protein degradation, mRNA degradation occurs at the M-G1 phase transition, and provides an important contribution to the reset of the transcriptome after cell division” changed to “Together, our findings demonstrate that, analogous to protein degradation, scheduled mRNA degradation occurs at the M-G1 transition. Scheduled mRNA degradation *likely* provides an important contribution to the reset of the transcriptome after cell division”.

Discussion: Page 14, lines 10-11

“Therefore, decay-mediated clearance of mRNAs as cells exit mitosis will aid to limit expression of the encoded proteins in G1 phase” changed to “Therefore, decay-mediated clearance of mRNAs as cells exit mitosis could aid to limit expression of the encoded proteins in G1 phase”.

Discussion: Page 14, lines 13-16

“As these genes include many genes that encode for proteins with important functions in cell cycle control, their continued expression in G1 phase may perturb normal cell cycle progression…” changed to “As these genes include many genes that encode for proteins with important functions in cell cycle control, we speculate that their continued expression in G1 phase may perturb normal cell cycle progression…”

CNOT1 contributes to the observed mRNA decay after mitosis• The data supporting this conclusion are relatively weak. The change in mRNA decay upon CNOT1 knockdown are all relatively minor. While the changes seem to be statistically significant, it’s not clear if these changes are large enough to be biologically relevant. They also don't appear robustly repeatable as CDK1 mRNA is rescued in Figure 4A but not in 4D.

We agree with the reviewer that he change in mRNA decay upon CNOT1 knockdown is modest. In accordance with this, we have softened our statements regarding the involvement of CNOT1 in mRNA decay (see below).

We do note that CDK1 levels in Figure 4D are lower upon knockdown of CNOT1, as is the case in Figure 4A (note that Figure 4D is plotted on a log scale, while Figure 4A is plotted on a linear scale). It is true that the effect of CNOT1 depletion is slightly stronger in Figure 4A than in Figure 4D (causing the effects in Figure 4D to be not statistically significant), but this is likely caused by the higher level of knockdown of CNOT1 by siRNA (Figure 4A) compared to CRISPRi (Figure 4D).

• I think the authors should edit the text here to reflect that CNOT1 may contribute somewhat to the decay but that it is not required, as requested by Reviewer 4, and to discuss additional factors that may be involved as requested by Reviewer 3.

As suggested by the reviewer we have weakened our statements regarding the involvement of CNOT1 in mRNA decay during the M-G1 phase transition, at several locations in the manuscript.

Introduction: page 4, line 33

“For several of these genes, we show that mRNA decay requires CNOT1” changed to “For several of these genes, we show that mRNA decay is stimulated by CNOT1”.

Main text, page 12, lines 3 to 4

“These results suggest that CNOT1-dependent deadenylation is important for mRNA decay at the M-G1 phase transition” changed to “These results suggest that CNOT1-dependent deadenylation is involved in mRNA decay at the M-G1 phase transition”.

Main text, page 12, line 14 to 15

“Collectively, these data show that CNOT1 is important for the decay of TOP2A and CDK1 mRNAs during the M-G1 phase transition […]” changed to ““Collectively, these data show that CNOT1 aids the decay of TOP2A and CDK1 mRNAs during the M-G1 phase transition […]”.

Also, we now discuss additional factors that could be involved in mRNA decay during the M-G1 phase transition in the discussion. On page 15, lines 5 to 8, we write:

“Nonetheless, it is possible that additional mechanisms contribute to mRNA decay during the M-G1 phase transition. Such mechanisms could involve PARN-dependent deadenylation, or may be independent of mitotic deadenylation and instead rely on mRNA decapping or endonucleolytic cleavage of the mRNA.”

In summary, I think the authors have either already addressed or have proposed experiments/analysis to address most of the reviewers comments about the methodology as well as the two waves of mRNA decay. However, the authors do not appear to have a clear plan to try and address the mechanism of the mRNA decay and whether or not CNOT1 is a major regulator of this decay. I think the relatively weak phenotypes with CNOT1 depletion are a major weakness of the manuscript but I recognize that the experiments needed to support their claims may not be feasible in a reasonable time.

We agree with the reviewer that it will be very interesting to perform additional work to provide a more detailed mechanism of mRNA decay during the M-G1 transition. However, as the reviewer indicates, this would require a significant amount of additional work, so we feel it is beyond the scope of the current manuscript.

Reviewer #3:I think that the paper by Tanenbaum and colleagues is suitable for publication in eLife as long as they:1) Perform the new experiments that they described in the rebuttal letter. Particularly important is to perform measurements of mRNA synthesis by addressing nascent transcription rates. The authors propose to include an analysis addressing the relative transcription levels of 'immediate decay' and 'delayed decay' genes. I agree with the authors that these results could clarify most of the concerns in this respect.

We have now included analysis of mRNA synthesis rates for *immediate* and *delayed decay* genes (see Figure 3A and Figure 3—figure supplement 1A). These results indicate that G1 phase transcription shutdown occurs for the large majority of genes that undergo scheduled mRNA decay, demonstrating a dual mechanism of gene silencing for these genes. These data are discussed in the main text on page 10, lines 2 to 9.

2) Carefully revise the text to avoid too strong conclusions not directly related to the results, as pointed out by the fourth referee.

As suggested by the reviewer we have weakened our statements regarding the involvement of CNOT1 in mRNA decay during the M-G1 phase transition, at several locations in the manuscript.

Introduction: page 4, line 33

“For several of these genes, we show that mRNA decay requires CNOT1” changed to “For several of these genes, we show that mRNA decay is stimulated by CNOT1”.

Main text, page 12, lines 3 to 4

“These results suggest that CNOT1-dependent deadenylation is important for mRNA decay at the M-G1 phase transition” changed to “These results suggest that CNOT1-dependent deadenylation is involved in mRNA decay at the M-G1 phase transition”.

Main text, page 12, line 14 to 15

“Collectively, these data show that CNOT1 is important for the decay of TOP2A and CDK1 mRNAs during the M-G1 phase transition […]” changed to ““Collectively, these data show that CNOT1 aids the decay of TOP2A and CDK1 mRNAs during the M-G1 phase transition […]”.

In addition, we have toned down our conclusions regarding the biological relevance of RNA decay after mitosis. We have done this at the following locations in the text:

Introduction: Page 4, line 36 to page 5, line 2

“Together, our findings demonstrate that, analogous to protein degradation, mRNA degradation occurs at the M-G1 phase transition, and provides an important contribution to the reset of the transcriptome after cell division” changed to “Together, our findings demonstrate that, analogous to protein degradation, scheduled mRNA degradation occurs at the M-G1 transition. Scheduled mRNA degradation *likely* provides an important contribution to the reset of the transcriptome after cell division”.

Discussion: Page 14, lines 10-11

**“**Therefore, decay-mediated clearance of mRNAs as cells exit mitosis will aid to limit expression of the encoded proteins in G1 phase” changed to “Therefore, decay-mediated clearance of mRNAs as cells exit mitosis could aid to limit expression of the encoded proteins in G1 phase”.

Discussion: Page 14, lines 13-16

“As these genes include many genes that encode for proteins with important functions in cell cycle control, their continued expression in G1 phase may perturb normal cell cycle progression…” changed to “As these genes include many genes that encode for proteins with important functions in cell cycle control, we speculate that their continued expression in G1 phase may perturb normal cell cycle progression…”.

3) Modify the Discussion section in order to consider the possibility that other mRNA decay factors may act in parallel to CNOT1 or in concert with it. This is particularly convenient if we considered the information supplied by the authors in response to Reviewer 1, describing the results of experiments addressing the possible involvement of DCP2 in post-mitotic mRNA decay (Figure 1A,B of the rebuttal letter).

In the Discussion section we now mention several other potential mRNA decay mechanisms that may contribute to cell cycle dependent RNA decay. On page 15, lines 5 to 8, we write:

“Nonetheless, it is possible that additional mechanisms contribute to mRNA decay during the M-G1 phase transition. Such mechanisms could involve PARN-dependent deadenylation, or may be independent of mitotic deadenylation and instead rely on mRNA decapping or endonucleolytic cleavage of the mRNA.”

In conclusion, I consider that this manuscript, with the new experiments and changes proposed by the authors, is a significant contribution on how mRNA stability regulation takes place accross the cell cycle, particularly after mitosis. This results should be of interest for a broad audience in the molecular and cell biology community, which would appreciate a paper like this in eLife.Reviewer #4:[…] The authors claim that their method can accurately measure the cell cycle stage and the "absolute" time an individual cell spent in the cell cycle with a resolution in the scale of minutes (p. 5). I have few concerns here, that affect also the strength of the biological conclusions. Here my specific points:a) The time resolution estimate is based on an average cell cycle length determined by fluorescence microscopy (Figure 1A). The cell-to-cell variability in cell cycle length should be computed and used to estimate the error of the time resolution.

The reviewer is right that cell-to-cell heterogeneity is important to consider in calculating an accurate cell cycle time. We note though that our cell cycle time values are not based on (average) cell cycle length, but rather on the FUCCI-G1 fluorescence intensities, which are dependent on the time since the previous cell division, rather than on the overall cell cycle duration. The FUCCI-G1 fluorescence intensity continues to increase the longer a cell spends in G1 phase, at least for the first 7 hours of G1 phase, and reports directly on the time a cell has spent in G1 phase, irrespective of that cell’s overall cell cycle duration. Times beyond 7 hours spent in G1 phase are more challenging, as some cells will enter S phase after 7 hrs of G1 phase, at which time the FUCCI-G1 reporter no longer accurately reports on the time in G1 phase. However, our analysis was limited to the first 7 hours of G1 phase, thus circumventing this issue.

b) The authors combined microscopy and FACS to calibrate the fluorescence measured with the two methods. In particular they base their calibration on the identification of cells in early S phase to normalize the two fluorescence datasets. Thus, accurate identification of early S phase cells seems very important to compare the fluorescence detected with the two methods. It is unclear from the text and methods section whether they used also independent approaches to determine the cell cycle phase. For instance in the FUCCI paper (Sakaue-Sawano et al. 2008 PMID: 18267078) Hoechst staining is used to independently determine the cell cycle. Furthermore, by microscopy, how did the authors determine when cells were undergoing early S-phase? Did they use FUCCI alone? To clarify this point, they could add a fluorescence microscopy video where they indicate how they classify cell cycle stages. This will be very important to ensure reproducibility of the method.

Per reviewers’ suggestion, and to improve reproducibility of our method, we have now included a video (Video 1), as well as still images of RPE-FUCCI cells to indicate the cell cycle classification used in this study (new Figure 1B). These have been added as figure references in the section describing the FUCCI system in the main text (page 5, lines 13-18).

Also, we have now included FACS-based analysis of DNA content of RPE-FUCCI. This analysis confirmed the gating strategy we used to identify G1 phase, early S, rest of S and G2/M phase cells (new Figure 1—figure supplement 1A) and is incorporated in our introduction of the FUCCI system in the main text (page 5, lines 15 to 30).

Finally, a detailed description for the criteria used to identify early S phase cells based on microscopy data can be found in the Methods section entitled “*Cell cycle timing using the FUCCI system”*, page 32, lines 9 to 23.

c) The FUCCI reporter expression heterogeneity is not clearly discussed in the text and it is unclear whether it may affect the accuracy of the method. The authors should provide a sample microscopy video (same as in (b) to show how heterogeneous (or not) is the expression of the reporter across many synchronized cells.

We agree with the reviewer that this in an important point. We have now included additional analysis of the heterogeneity of FUCCI-G1 fluorescence and have determined the corresponding error in the G1 timing of single cells (see new Figure 1—figure supplement 1C), which we discuss in the main text (page 5 lines 28 to 31) and in the discussion (page 13, lines 21 to 24). In addition, we have added a video of many cells expressing the FUCCI reporters (Video 1), as suggested by the reviewer.

d) The biological discovery of this paper is that mRNA levels decline in multiple waves during mitotic exit. I find the text at times confusing. For instance at page 7, the paragraph title "mRNA levels decline in multiple waves after cell division", suggests that both degradation waves occur after completion of cell division. But this is not what is shown in Figure 2D. The "immediate" degraded mRNAs are degraded before the end of mitosis, according to the FISH quantifications. I highly recommend to clarify the text.

The reviewer is correct that this description is correct. We apologize for this inaccuracy and we have modified the title of our manuscript and the title of the relevant paragraph to more accurately reflect the data presented.

Old title: “Time-resolved single-cell sequencing identifies multiple waves of mRNA decay during mitotic exit” changed to “Time-resolved single-cell sequencing identifies multiple waves of mRNA decay during the mitosis-to-G1 phase transition”.

Paragraph title (page7): “mRNA levels decline in multiple waves after cell division” changed to “mRNA levels decline in multiple waves during the M-G1 transition”.

e) The method presented in this manuscript provides the ability to precisely estimate the duration of the different cell cycle phases of HEK293 cells. One result that I would recommend adding is the duration of the different cell cycle phase of an average HEK293 cell, based on your FUCCI reporter estimates. This will be important to interpret many of your plots where the X axis is presented as "Time after metaphase". I make this point because I think that some of the mRNA levels declines reported in your study may be explained by a dilution effect occurring upon cell division. I think that this is particularly important for the "delayed" mRNA pool. Could the authors please comment this point in the text? How did they consider dividing cells while performing single cell imaging?

We thank the reviewer for this suggestion. We have now included analysis of G1 phase length, based on FUCCI-G1 fluorescence, in both RPE-FUCCI wild-type as well as RPE-FUCCI p53-knock-out cells (Figure 2—figure supplement 1K). These data show that the average G1 phase length is ~12 hours, and the shortest G1-phase durations are ˜6-7 hours. The entire cell cycle duration in RPE cells is ~24 hrs. So within the time-window of our analysis (~7 hours post metaphase), almost no cells with have entered the next S-phase and no cells will have passed through *another* mitosis. Therefore, the reported mRNA decline is not affected by the next cell division. Additionally, it is important to note that using our FACS-based isolation of cells, we only include cells that are still in G1 phase (see sorting strategy in Figure 1C), ensuring that progression into the next cell cycle phase(s) (e.g. S-phase) does not complicate our analysis.

Regarding the dilution effect occurring upon cell division (i.e. mRNAs are divided over the two daughter cells), specifically when comparing G2/M sequenced cells to early G1 phase cells; in this case there is indeed a “dilution effect”, which we have corrected in all our experimental approaches. In our qPCR-based analysis we always normalize gene expression to two independent control genes which were previously reported to have long half-lives (Ribophorin and GAPDH), using the ∆∆Ct-method. Since these control mRNAs will also be diluted as a consequence of cell division, normalization of gene expression to these two control genes will compensate for any dilution of mRNAs caused by division. As such, we believe that the reported decrease in mRNA expression is due to turn over rather than due to dilution. In the sc-RNA sequencing analysis, we have normalized all cells for the number of UMIs retrieved (using an algorithm from Monocle2). In this way we compensate for the dilution effect and we only look at the effect of mRNA decay. We have updated the methods section to more clearly explain this important point (page 29, line 32 to page 30 line 2, and page 31, lines 12 to 13).

Finally, for our FISH data (Figures 2E, 2F and 4A) where we imaged mRNA content in single cells, we always included both daughter cells for our analysis of mRNA content, as this prevents a dilution effect that is caused by cell division itself. With regards to FUCCI-G1 fluorescence quantification, we analyzed FUCCI-G1 fluorescence levels in both daughter cells individually, as this is also how FUCCI-G1 fluorescence will be analyzed using FACS.

f) FISH is used as in independent mean to measure mRNA reduction in single cells at different stages of the cell cycle. For these experiments, two mRNAs belonging to the "immediate" degraded group are measured, TOP2A and CDK1 and both nicely show a decrease before the end of the cell cycle. Could you add the single cell data to the FISH quantifications shown in figures 2E, 2F, and 4A, to report the cell-to-cell variability observed for the expression of these mRNA?

We have now included the single cell data of a representative experiment corresponding to the data shown in figures 2E and 2F (new Figure 2—figure supplement 1B-C), and corresponding to figure 4A (new Figure 4—figure supplement 1C-D).

g) As a follow-up question, why didn't you perform FISH also on two mRNAs from the "delayed decrease" group, to show changes in mRNA expression? This seems an important control if the aim is to demonstrated trough impendent methods that two waves of degradation occur during the M/G1 phase transition.

We did not include any genes belonging to the *delayed* decay group in our smFISH experiments as we had already validated the existence of two independent waves of mRNA decay using RT-qPCR (Figure 2C.) (and due to the high price of smFISH probes). The smFISH experiment was performed to more precisely determine the moment of mRNA decay for genes belonging to the *immediate decay* group. We have now included additional experiments investigating the mRNA levels of 5 *immediate decay* genes and 5 *delayed decay* genes in cells FACS-sorted in G2, early and late mitosis. These results confirm that *immediate decay* genes start to decline in mitosis, while *delayed decay* genes remain stable during mitosis (new Figure 2—figure supplement 1E-H).

h) The authors conclude that "mRNA degradation occurs at the M/G1 phase transition, and provides an important contribution to the reset of the transcriptome after cell division". However, no experiment is performed to determine the importance of this decay mechanism and its consequences on cell cycle. For instance it could be tested by selectively downregulating the expression of CNOT during the M/G1 phase transition by using an inducible degron system. I understand that this may be a complicated experiment to do, but if a demonstration is not provided, then the text should remain more conservative in describing the results.

As suggested, we have toned down our conclusions regarding the biological relevance of RNA decay after mitosis. We have done this at the following locations in the text:

Introduction: Page 4, line 36 to page 5, line 2

“Together, our findings demonstrate that, analogous to protein degradation, mRNA degradation occurs at the M-G1 phase transition, and provides an important contribution to the reset of the transcriptome after cell division” changed to “Together, our findings demonstrate that, analogous to protein degradation, scheduled mRNA degradation occurs at the M-G1 transition. Scheduled mRNA degradation *likely* provides an important contribution to the reset of the transcriptome after cell division”.

Discussion: Page 14, lines 10-11

“Therefore, decay-mediated clearance of mRNAs as cells exit mitosis will aid to limit expression of the encoded proteins in G1 phase” changed to “Therefore, decay-mediated clearance of mRNAs as cells exit mitosis could aid to limit expression of the encoded proteins in G1 phase”.

Discussion: Page 14, lines 13-16

“As these genes include many genes that encode for proteins with important functions in cell cycle control, their continued expression in G1 phase may perturb normal cell cycle progression…” changed to “As these genes include many genes that encode for proteins with important functions in cell cycle control, we speculate that their continued expression in G1 phase may perturb normal cell cycle progression…”.

We do note that we only state that mRNA decay may affect the transcriptome (i.e. mRNA levels), not that mRNA decay has functional consequences for the cell cycle.

i) The authors conclude that "mRNA decay requires CNOT1". The CNOT1 downregulation experiments show that CNOT1 contributes to decay, not that it is required. This should be changed in the text.

As suggested by the reviewer we have weakened our statements regarding the involvement of CNOT1 in mRNA decay during the M-G1 phase transition, at several locations in the manuscript.

Introduction: page 4, line 33

“For several of these genes, we show that mRNA decay requires CNOT1” changed to “For several of these genes, we show that mRNA decay is stimulated by CNOT1”.

Main text, page 12, lines 3 to 4

“These results suggest that CNOT1-dependent deadenylation is important for mRNA decay at the M-G1 phase transition” changed to “These results suggest that CNOT1-dependent deadenylation is involved in mRNA decay at the M-G1 phase transition”.

Main text, page 12, line 14 to 15

“Collectively, these data show that CNOT1 is important for the decay of TOP2A and CDK1 mRNAs during the M-G1 phase transition […]” changed to ““Collectively, these data show that CNOT1 aids the decay of TOP2A and CDK1 mRNAs during the M-G1 phase transition […]”.

[Editors' note: we include below the reviews that the authors received from Review Commons, along with the authors’ responses.]

Reviewer #1 (Evidence, reproducibility and clarity (Required)):In the manuscript "Time-resolved single-cell sequencing identifies multiple waves of mRNA decay during mitotic exit", Krenning et al. describe a method that connects live-cell microscopy with single-cell RNA sequencing in order to monitor global changes in mammalian mRNA gene expression in a cell-cycle-dependent manner. To this end they employ a fluorescent, ubiquitination-based cell cycle indicator (FUCCI system) in human untransformed RPE-1 cells coupled to SORT-Seq in order to generate a high-resolution, time-resolved transcriptome profile of cells spanning the transition form M phase into G1 phase. By comparing FACS-based sampling with an in silico trajectory inference method the authors provide convincing evidence that this system allows for high-resolution, time-resolved transcriptome profiling of cells in the transition from M-phase into G1 phase. Subsequent analysis of changes in steady-state gene expression revealed a set of >200 transcripts that undergo rapid decay in two distinct waves, first around the time of mitotic exit and second upon G1 entry. Those results were independently validated by single molecule FISH of select transcripts, revealing that the first wave of RNA decay initiates at the start of anaphase and the second wave starts during early G1 phase. Using mathematical modeling followed by selective validation (using cell-cycle staging combined with global inhibition of RNA decay), the authors derive and confirm precise mRNA decay rates for cell cycle regulated mRNAs that were overall lower than what has been observed in previously described population-based half-life measurements. Finally, the authors establish a role of deadenylation by the CCR4-NOT complex in selective mRNA decay during M-phase/G1 transition by revealing a partial stabilization upon NOT1-depletion by RNAi.I have only a minor comment to help improve the mechanistic insights: While the authors attribute the partial stabilization of cell-cycle regulated mRNAs to an incomplete depletion of NOT1 (which is possible), an alternative hypothesis would be that decapping and 5´-to-3´ decay further contribute to mRNA turnover. Notably, the authors describe the use of siRNAs targeting Dcp2 in the methods section (without referring to it in the results part), indicating that they may have data that could clarify this. They authors may consider adding this data to complete an otherwise nicely executed and well-written manuscript.

We agree with the reviewer that decapping and 5’-to-3’ decay could contribute to mRNA turnover, either following initial CNOT1-dependent deadenylation or in the absence of deadenylation. Indeed, we had perform studies addressing the involvement of DCP2 in the decay of CDK1 and TOP2a mRNAs during mitosis, but we decided to omit these results, as they are largely inconclusive. We found a small but significant reduction in decay of TOP2A, and a small, but not significant reduction in the decay of CDK1. As we depleted DCP2 by siRNA, it is difficult to interpret these results, as the lack of effect could be explained by insufficient knockdown. We have included these experiments as Author response image 1,B.

**Author response image 1. sa2fig1:** Figure 1. (A) Cells were transfected with the indicated siRNAs and fixed 48 hours later. mRNAs, DNA and membranes were labeled with smFISH probes, DAPI and fluorescent wheat germ agglutinin, respectively. mRNA numbers of cells in late mitosis were counted and divided by the number of mRNAs present in cells in early mitosis. (B) Cells were transfected with the indicated siRNAs. 48 hours later cells were lysed and the RNA was harvested. Gene-expression levels were analyzed by qPCR, relative to cells treated with siRNAs targeting Luciferase.

Due to the inconclusive nature of these experiments, we propose to omit these data from the manuscript. Instead, we will include additional discussion on the possible involvement of other decay factors, either acting in parallel to CNOT1 or in concert with CNOT1, in the Discussion section. We will make the following changes:

– Remove the DCP2 data from Figure S4A.

– Change the figure legends for Figure S4A accordingly.

– Remove the siRNA from the materials section.

– Remove the DCP2 data in Supplemental Table 2.

– Include discussion on the possible involvement of other decay factors.

Reviewer #1 (Significance (Required)):Overall, this is an impressive and well-controlled body of work providing novel insights into cell-cycle controlled mRNA decay. It reveals novel insights into the targets and molecular mechanisms of mRNA decay in the transition of mitosis to G1, thereby adding significantly to our understanding of the scope of tightly controlled gene expression for proper execution of the cell cycle.Reviewer #2 (Evidence, reproducibility and clarity (Required)):Summary:In this study Krenning, Sonneveld, and Tanenbaum have investigated the temporal control of mRNA decay after mitosis. Previous work has demonstrated that following mitosis, regulated protein degradation serves an important role in erasing any lingering memory of the previous cell cycle. Left unanswered however, is whether mRNA decay is similarly regulated after mitosis, and if so, what role does it play in the cell cycle. The authors use time-lapse imaging and single-cell RNAseq to measure decay rates of mRNAs after mitosis. They use the FUCCI sensors to identify the precise age of each individual cell prior to performing single-cell RNAseq. Using this pseudo-timelapse approach, they identify several genes that are actively degraded after mitosis. Interestingly, they find that some mRNAs are decayed rapidly and immediately after mitosis while other mRNAs are decayed only after a delay of about 1.5 hours after mitosis. The authors find that, at least for several of the genes, the protein CNOT1 mediates the decay. The authors conclude that regulated mRNA decay occurs after mitosis and helps reset the transcriptome at the onset of a new cell cycle.The question being asked in this manuscript is interesting and potentially very important to the cell cycle field. However, I believe this study suffers from several technical issues highlighted below as well as relatively modest effect sizes, particularly when it comes to ascribing CNOT1 as the key regulator mediating mRNA decay upon mitotic exit.Major Comments:1) The authors repeatedly claim that they can accurately measure the time a cell has spent in G1 phase, due to using live-cell imaging to calibrate their FACS data. However, following mitosis, cells bifurcate into either G0/Quiescence or into G1 phase. Importantly, the FUCCI-G1 sensor is not capable of distinguishing between these two cell cycle phases. Thus, while two cells may have spent the same amount of time since mitosis, one cell may be in Quiescence while the other cell is in G1 phase and they would both have the same level of FUCCI-G1. Even by live-cell imaging it’s not possible to distinguish between these two cell cycle phases with the markers used in this study. I believe you can see evidence for this in the Flow cytometry data in Figure 1C where there is a large population of cells with very high FUCCI-G1 levels (even higher than most of the S phase cells) but very low FUCCI-G2 levels. These cells likely represent cells that entered G0 after mitosis, remained there for several hours, and thus accumulated very high levels of the FUCCI-G1 sensor. This technical limitation has several implications. First, it means what is likely going into the single-cell mRNA seq workflow is a mixture of G0 and G1 cells. Since the authors observe two distinct "waves" of mRNA decay, it makes me wonder if one wave might be occurring in G1 cells while the other wave is occurring in G0 cells?Second, the authors state several times they can accurately determine the time a cell has spent in G1 phase. The more accurate statement however is that the authors can accurately determine the time that has elapsed since mitosis, since they cannot distinguish between G0 and G1 phase. This is admittedly a nuanced distinction but an important one given that quiescence cells are in a very different cellular state than cells in G1 phase. The authors actually use the correct x-axis label of "Time after metaphase (min)" in several of their figures, but they do not use the same language in the main text or to describe their conclusions. To overcome the technical challenge of distinguishing between G0/G1 cells and to address these two points, the authors could use an additional FUCCI sensor, mVenus-p27K(-), which accumulates in G0 cells but not in G1 cells (PMID: 24500246).

We thank the reviewer for pointing out the G0/G1 bifurcation that occurs after metaphase, and agree that based on the FUCCI-G1 reporter we cannot distinguish between cells entering G0 or G1 phase. The reviewer is also correct that, in principle, it is possible that one wave of mRNA decay occurs specifically in cells entering G1 phase, while the other occurs in cells entering G0 phase. To address this point we have performed additional analyses and we propose to perform an additional experiment as well. In addition, we will make textual changes to the manuscript, addressing the G0/G1 bifurcation.

As the reviewer suggests, it could be possible that one wave of mRNA decay occurs in G1 cells, while the other wave occurs in G0 cells. We have performed new analysis comparing genes that are differentially expressed (DE) between G0 and G1 phases (genes were selected based on the study mentioned by the reviewer, PMID 24500246) to genes that are decayed following cell division. We found that many genes that are subject to post-mitotic decay are also differentially expressed between G0/G1 phases. Of the genes belonging to the ‘immediate decay’ group, 57% is also differentially expressed between G1 and G0. Of the genes belonging to the ‘delayed decay’ group, 73% is differentially expressed between G1 and G0. This implies that genes that are differentially expressed between G0 and G1 phases can be decayed during both waves of mRNA decay. There results argue against one wave of mRNA decay occurring in G1 cells and the other in G0 cells.

Additionally, if the two decay waves would occur in two distinct sub-populations of cells (i.e. G1 and G0 cells), it is expected that cells that show the strongest decay in one set of genes (e.g. immediate decay genes) would show no, or substantially less, decay of the other set of genes (e.g. delayed decay genes). In other words, the levels of immediate and delayed decay genes should *anti*-correlate. In contrast, if both waves of mRNA decay occur sequentially in the same cells, irrespective of whether cells are/become G1 or G0 cells, such anti-correlation in the levels of immediate and delayed decay genes is not expected. We have now performed this analysis and found that cells with low levels of immediate decay genes, also tend to have low levels of delayed decay genes (Author response image 2) . The expected anti-correlation is not observed, rather our results show that both waves of mRNA decay occur in the same cells, indicating that these two waves of decay do not reflect decay events that occur selectively either in G0 and G1 phase.

**Author response image 2. sa2fig2:** Comparison of immediate decay versus delayed decay in single cells. We generated two metagenes, one including all genes belonging to the ‘immediate decay’ wave, and one including all genes belonging to the ‘delayed decay’ wave. Metagenes were created by summing up all reads for the genes that belonging to the same wave of decay in a single cell, thus creating one expression value per metagene per cell. Then we analyzed metagene expression at 240 minutes post metaphase normalized to its expression at G2/M.

While we found a good correlation in the decay of genes belonging to both decay waves, we wanted to further investigate if we could detect evidence for a bifurcation of G1 and G0 cells on the whole transcriptome level. For this, we looked back at our Monocle2 analysis, which generated a transcriptome-based trajectory. In case G1 and G0 cells would differ on the transcriptome level, we would expect to see a bifurcation in the trajectory at some point during G1 phase. We did not detect a bifurcation in cells during the first 4 hours after cell division (Figure 1-figure supplement 1M). These observations show that, even though the cells are a mix of cells in G1 and G0, or a mix of cells that will enter G0 or G1, the transcriptomes of all cells (and thus the mRNA decay) does not yet show signs of the G0-G1 bifurcation during the first 4 hours following cell division (the time-window used to identify the two waves of mRNA decay).

Lastly, to further explore this point experimentally, we propose the following experiment; we will generate p53 knockout cells, as the quiescence/proliferation decision is dependent on p53 and its transcriptional target p21 (Spencer et al., 2013; Yang et al., 2017). We will perform new qPCR-based analysis, directly comparing the mRNA decay of populations of wild-type cells (containing both G1 and G0 cells) to p53 knockout cells (containing only G1 cells). This will experimentally validate whether the lack of G0 cells prevents the mRNA decay for either group of genes.

2) One of the main novelty claims of the paper is that mRNA decays following mitotic exit in two waves. In order to support this claim, the authors plot the time since metaphase vs normalized transcripts for many single cells. For several genes including CDK1, TOP2A, and UBE2C, there is an immediate drop from time 0 to the next most densely populated timepoint of about 20 minutes. This results in a bimodal histogram as seen in Figure 2A, where there is one single bin representing genes that are maximally decayed at time 0, while another population is more normally distributed with a median decay time of ~80minutes. I have two concerns about this data. First, there are very few if not zero cells analyzed that were between 0 and 20 minutes old at the time of collection. Therefore the authors do not know if the mRNA decayed immediately after mitosis or with a 20minute delay after mitosis. Thus, rather than some genes decaying immediately after mitotic exit resulting in that one bin at time=0 and some genes decaying with a delay, there could actually be just one wave of mRNA decay that is broadly and normally distributed from 20-80minutes after mitosis (see histogram Figure 2 and Figure S3G; if you ignore anything less than 20minutes due to lack of data during this time window, then there is just a single distribution). While likely technically challenging due to cytokinesis, sampling cells that are between 0-20minutes old may be important to accurately measuring the decay rates of mRNAs upon mitotic exit.

We apologize for the confusion caused by this data. While it is true that only few data points exist between 0 and 20 min post-metaphase, the data nonetheless shows that the first wave of decay initiates before the start of G1. We will modify the text to explain this point more clearly (see below for a detailed explanation) and we will perform an additional experiment to further strengthen this point.

To determine the moment of initiation of decay, we performed a data fitting approach in which we fit the mRNA levels over time of each gene with an exponential decay distribution with a variable delay time (i.e. an initial plateau phase followed by the exponential decay). We then searched for the delay time (in intervals of 10 minutes) and the decay rate that best fit the data. This analysis revealed that for many genes a delay time of 0 min (relative to metaphase) generated the best fit. It is perhaps counterintuitive that a delay time of 0 minutes could create the best fit for a dataset that includes very little data points in the first 20 min of the analysis. Yet, this is possible because one can fit the later timepoints of an exponential decay distribution and based on the fit of the later datapoints extrapolate the earlier datapoints. This approach allowed us to model the decay for each gene, determining the best starting point of the exponential decay distribution. The modeling approach could have determined decay to initiate at 0, 10, 20, 30, 40 (etc) minutes post metaphase as the best fit. However, for 75% of the ‘immediate decay genes’ 0 minutes post metaphase was identified as the best fit, indicating that decay of the immediate decay genes is initiated quite synchronously during mitotic exit.

Importantly, we experimentally validated that decay initiates during mitotic exit for two genes that were predicted to initiate decay at 0 minutes post metaphase, CDK1 and TOP2A, using smFISH (see manuscript Figure 2D-F). These data show that declining mRNA levels of CDK1 and TOP2a are detectable as early as anaphase, which independently validates (the end of) mitosis as the moment of mRNA decay initiation.

To further corroborate these findings, we propose to include smFISH data for an additional gene that is predicted to initiate decay at 0 minutes post metaphase FBXO5. In addition, we propose to separate cells in telophase from prometaphase/metaphase cells and also from cells in G2 (based on mitotic shake-off and FACS sorting) and assess the mRNA levels of 5 immediate decay genes using qPCR in these different phases. These experiments will hopefully confirm that decay indeed already initiates in anaphase/telophase for additional genes.

Second, it’s not entirely clear from the methods section, but I am assuming that for time 0 of the plots in Figure S3A-F, the authors are using G2/M sorted cells as defined by FUCCI-G2 high/FUCCI-G1 low status. However, depending on what precise FACS gates were used to do this sort, these cells could be anywhere in early G2, late G2, prometaphase, metaphase, etc and they are all averaged together. Thus, when the authors claim that the mRNAs are decaying immediately after mitosis, that's based on comparing the mRNA levels in this mixed G2/M population with cells that are 20minutes or more after mitosis. For genes like CDK1 where the mRNA is almost completely gone at 20 minutes after mitosis when they first have any cells to measure, its entirely possible these mRNAs were already decayed either before or within mitosis, rather than upon mitotic exit. What we really need to know is what are the mRNA levels precisely at metaphase and plot that value as time=0 (for example in figure S3A-F). Perhaps the authors could use CDK1i plus Taxol treatment to accumulate cells in mitosis like they describe in the methods for the transcription inhibition experiments. (If this is what was done for the Sort-Seq experiments as well as the data in Figure S3A-F, my apologies for not completely understanding, but the methods section should be updated to make this more clear)

The reviewer is correct that the cells represented at t=0 are a mix of G2 cells and early mitotic cells. While the concern that mRNA decay already initiates during G2 or early mitosis is understandable, there are several experiments that demonstrate that mRNA decay does not initiate until post-metaphase.

1. Our previous mRNA sequencing experiments has directly compared mRNA levels in G2 vs early mitosis, which revealed that there are little to no changes in mRNA expression between G2 and mitosis for the genes that undergo mRNA decay post-metaphase (Tanenbaum et al., *eLife*, 2015).

2. Using smFISH, we could show that the levels of two genes identified in this study, CDK1 and TOP2a, are stable during mitosis, until cells enter anaphase.

To further support the conclusion that mRNA decay initiates post-metaphase, and mRNA levels between G2 and early mitosis remain largely unchanged we will perform the experiment proposed above (separating G2 cells from early mitotic cells and telophase cells) and examine mRNA levels at these different time-points. This will allow us to experimentally determine the putative changes in gene expression between G2, early mitosis and post metaphase.

3) As the authors point out in the discussion, the effect size of CNOT1 depletion on mRNA decay are rather small. Notably the genes that appear to be statistically significant are all the genes that are hardly decayed after mitosis anyways (see Figure 4D, ARLGIP1, PSD3, etc). These genes appear to be only down about 2-fold (-1 on a log2 scale axis) in the control group, which would be expected during mitosis when the mother cell splits in half into two daughter cells. Depending on the method of normalization, this change in mRNA levels may simply be due to this. I understand the technical limitations in knocking down an essential gene, but given that one of the major novelty claims of this study is that CNOT1 mediates the mRNA decay upon mitotic exit, more robust proof is warranted. The authors could employ the inducible protein degradation system they referenced in the discussion. Additionally they could perhaps show data that CNOT1 is somehow differentially regulated during mitotic exit that would account for this sudden change in mRNA stability. If the authors were to include additional corroboratory data demonstrating CNOT1 is most active during mitotic exit for example, it would help to overcome that low effect size upon mild CNOT1 depletion.

The statement that depletion of CNOT1 only significantly prevents mRNA decay for genes that are hardly decayed after mitosis is not accurate. CNOT1 knockdown significantly inhibited the decay of CDK1 and TOP2a mRNAs (manuscript Figure 4A), and of UBE2C and FZR1 (manuscript Figure 4D). These are all genes that show a very strong mRNA post metaphase. In fact, for all the genes tested, CNOT1 depletion reduced post-metaphase decay for 9 out of 10 genes.

The reviewer correctly points out that, depending on the normalization method used for qPCRs, one may expect a 2-fold reduction in the absolute mRNA levels due to cell division itself. In our experiments, we compare gene expression of decayed genes to two independent control genes that are not degraded post mitosis. This comparison will correct for any effects on mRNA levels caused by cell division. We will update the methods section to clarify this important point.

Finally, the reviewer suggests several new lines of experimentation to further study the involvement of CNOT1 in mRNA decay during mitotic exit. While we agree that using degron-tagging of the endogenous CNOT1 locus and developing methods for activity profiling of CNOT1 would be very interesting and exciting, these approaches represent a significant investment in time and we feel that this work goes beyond the scope of the current study.

Minor comments:1) The authors state that they can "accurately determine the time a cell has spent in G1 phase based on its FUCCI-G1 fluorescence as measured by FACS" (Page 6, last line of the first paragraph). It would be nice to know how accurately? What is the level of uncertainty in your measurements? +/- how many minutes?

The reviewer raises an important point. The inference of timing of single cells is indeed subject to variability in FUCCI-G1 fluorescence. We will measure the experimental error in this inference and include it in the manuscript. It is important to note that mRNA decay rates are calculated based on many cells, averaging out any measurement error.

2) The authors only focused on those genes whose mRNA’s were decayed upon mitotic exit, but of equal interest would be those cell cycle genes that were not observed to decay upon mitotic exit. The authors may want to highlight these genes, because it might shed light on what aspects of the cell cycle cells want to reset, and what aspects of the cell cycle the cells may wish to “remember” from the previous cell cycle. For example, what is the status of the cyclins? CIPs? Origin licensing genes? There is growing evidence that the status of the mother cell can influence the cell “ycle of ”he daughter cell and mRNA levels of key cell cycle genes have been implicated in this (PMIDs: 28514656, 28317845, 28869970).

This is an interesting point. We have provided an easily-searchable excel sheet with an overview of all genes and their expression levels over time during the M-G1 transition (Supplementary file 1), so that anyone can examine the expression of their favorite gene.

Reviewer #2 (Significance (Required)):This paper offers a potential conceptual advance to our understanding of how cells reset after the cell cycle is completed. By demonstrating that cells regulate protein as well as mRNA degradation upon cell cycle exit, the authors show cells utilize multiple mechanisms to reset the biochemical state of the cell at the onset of a fresh cell cycle. This work would be of potential interest to the cell cycle field, particularly those interested in G1 regulation. It would also be of interest to people interested in mRNA regulation as it would demonstrate a clear window during which mRNA decay was specifically upregulated.While my expertise lies in the cell cycle and the use of FUCCI sensors to identify cell cycle stages, I have less expertise in single-cell RNAseq and methods of mRNA quantification.Reviewer #3 (Evidence, reproducibility and clarity (Required)):In this paper, Dr. Tanenbaum and colleagues present a new method to determine the exact point of the cell cycle in which a cell is. This new method has the potential to be applied to any cell-cycle phase and even to other processes. Focusing on M-G1 transition, they do sequencing of single human cells. They identify two groups of cell cycle-related genes, expressing mRNAs which are degraded either immediately after exit from mitosis or later on during G1. One of the factors involved in this degradation is identified as CNOT1.In my opinion, the new method is well stablished and has the potential to be very useful to future work. In general, I think that the main conclusions are based on more than one approaches and are convincing. In relation to these, I have two major concerns:1. Although it is clear that a scheduled mRNA decay exits, this does not exclude the possibility of a concomitant effect on mRNA synthesis. A measurement of nascent transcription is needed.

The reviewer raises an important point, and we believe that concomitant transcription shut down may indeed play a role. Therefore, we propose to include an analysis addressing the relative transcription levels of ‘immediate decay’ and ‘delayed decay’ genes.

2. As mentioned in the discussion, it is possible that the limited effect of depleting CNOT1 is due to the partial knockdown. However, it is also possible that a different pathway of mRNA degradation is involved. This should be addressed by targeting other decay factors (for example Xrn1 and/or an exosome component).

We agree with the reviewer, and we had attempted to test the involvement of other mRNA decay factors, including the decapping factor DCP2 in post-mitotic mRNA decay. We decided to omit these results, as they are largely inconclusive unfortunately. We found a small but significant reduction in decay of TOP2A, but no significant reduction in the decay of CDK1. As we depleted DCP2 by siRNA, it is difficult to interpret these results, as the lack of effect could be explained by insufficient knockdown. We have included these experiments here (Figure 1A,B, see response to reviewer 1). As these data are inconclusive, we propose to leave them out of the manuscript. Instead, we will include additional discussion on the possible involvement of other decay factors, either acting in parallel to CNOT1 or in concert with CNOT1, in the Discussion section.

Minor comments:1. Cell cycle control is not absolutely universal. The authors should mention that their results and conclusions correspond to human cells; if not in the title, at least in the abstract.

We will include this statement in the abstract.

2. Is SORT-seq (mentioned exclusively in the methods section) the same as scRNA-seq?

SORT-seq is a combination of FACS-sorting and scRNA-seq. We will explain this more carefully in the results and methods sections of the manuscript.

3. Figure 2: there is no correspondence between the TOP2A images and their quantification. This experiment also needs an unrelated mRNA FISH as a negative control.

We apologize for this mistake and will update the figure with a more representative image.

4. Figure 3D: if I correctly understood this experiment, “Time in Actinomycin D” is a better title for the X axis. “Time after mitotic shake-off” is misleading because it suggests that the cells were released from the mitotic blockage.

We thank the reviewer for pointing this out. We will update the figure accordingly.

5. Figure S3L: the blockage of transcription by Actinomicin D should be demonstrated by a more general method, such as EU (5-ethynyl uridine) incorporation.

Actinoymcin D is a very well-established inhibitor of RNA polymerases. We show through expression of a control gene (CDKN1a) that the drug is active in our experiments, so we feel that additional analysis is not essential in this case.

6. Figure 4A: what is the cell cycle phenotype of CNOT1 siRNA? If a reduction with CRISPRi of 50 % causes a phenotype (Figure S4C), then a reduction of 90 % by the siRNA (Figure S4A) should cause a stronger phenotype.

We agree with the reviewer that a higher knockdown efficiency is expected to more profoundly affect cell cycle progression. We propose to include the analysis of cell cycle progression upon siRNA mediated CNOT1 depletion in the revised manuscript.

7. Do the two waves of protein degradation (mentioned in the second paragraph of page 12) affect the same genes than the two waves of mRNA degradation?

This is an interesting point that we will address. We will provide a supplemental table containing an analysis comparing molecular targets of the APC/C during mitotic exit with the genes classified as ‘immediate decay’ or ‘delayed decay’.

Reviewer #3 (Significance (Required)):The new method described in this paper could be useful to address many questions in the field of cell-cycle control. Moreover, with modifications, it could be applied to any time-dependent biological problem. In addition, the scheduled degradation of mRNAs in the M-G1 transition is a discovery of biological significance. In my opinion, the paper is technically interesting to a broad range of investigators, and biologically relevant for those studying the cell cycle.Keywords of my field of expertise: transcription, chromatin, gene expression.

References:

Spencer, S. L., Cappell, S. D., Tsai, F., Overton, K. W., Wang, C. L., and Meyer, T. (2013). The Proliferation-Quiescence Decision Is Controlled by a Bifurcation in CDK2 Activity at Mitotic Exit. *Cell*, *155*(2), 369–383. https://doi.org/10.1016/j.cell.2013.08.062

Yang, H. W., Chung, M., Kudo, T., and Meyer, T. (2017). Competing memories of mitogen and p53 signalling control cell-cycle entry. *Nature*, *549*(7672), 404–408. https://doi.org/10.1038/nature23880